# Strain localisation and dynamic recrystallisation in the ice-air aggregate: A numerical study

Florian Steinbach[1,2], Paul D. Bons[1], Albert Griera[3], Daniela Jansen[2], Maria-Gema Llorens[1], Jens Roessiger[1], Ilka Weikusat[1,2]

[1]Department of Geosciences, Eberhard Karls University Tübingen, 72074 Tübingen, Germany
[2]Alfred Wegener Institute Helmholtz Centre for Polar and Marine Research, 27568 Bremerhaven, Germany
[3]Departament de Geologia, Universitat Autònoma de Barcelona, 08193 Bellaterra (Barcelona), Spain

*Correspondence to:* Florian Steinbach (florian.steinbach@uni-tuebingen.de)

**Abstract.** We performed numerical simulations on the micro-dynamics of ice with air inclusions as a second phase. Our aim was to investigate the rheological effects of air inclusions and explain the onset of dynamic recrystallisation in the permeable firn. The simulations employ a full field theory crystal plasticity code coupled to codes simulating dynamic recrystallisation processes and predict time-resolved microstructure evolution in terms of lattice orientations, strain distribution, grain sizes and grain boundary network. Results show heterogeneous deformation throughout the simulations and indicate the importance of strain localisation controlled by air inclusions. This strain localisation gives rise to locally increased energies that drive dynamic recrystallisation and induce heterogeneous microstructures that are coherent with natural firn microstructures from EPICA Dronning Maud Land ice coring site in Antarctica. We conclude that although overall strains and stresses in firn are low, strain localisation associated with locally increased strain energies can explain the occurrence of dynamic recrystallisation.

## 1 Introduction

The ice sheets on Greenland and Antarctica are composed of snow layers, originally containing a large proportion of air, which are transformed into solid ice due to compaction and sintering processes (Herron and Langway, 1980; Colbeck, 1983). At the firn-ice transition, the air is sealed off in bubbles as the pores are no longer connected and do not allow exchange with air from other layers or the atmosphere (Schwandner and Stauffer, 1984; Stauffer et al., 1985). For this reason, the ice sheets are considered as valuable archives of the paleo-atmosphere (Luethi et al., 2008; Fischer et al., 2008). However, ice sheets are not static, but flow under their own weight, which can potentially cause the paleo-climatic record to lose its integrity (Faria et al., 2010). For the interpretation of these records it is essential to not only understand the deformation dynamics of polycrystalline ice, but also the implications of a second phase in the form of air bubbles. For consistency, we use the expression *air bubbles* whenever referring to the natural material and *air inclusion* only for numerical models, where the individual units of air are neither interconnected, nor communicating.

Compared to other abundant minerals on the surface, ice on earth is always at high homologous temperatures ($T_h = T_{actual}/T_{meltpoint}$), close to its pressure melting point and therefore creeping under gravitational forces (Petrenko and Whitworth, 1999; Faria et al. 2014a). The macroscopic behaviour of the ice aggregate results from the local response of individual ice crystals and the distribution of second phases within the polycrystalline aggregate.

Deformation in the ice crystal is mainly accommodated by dislocations, meaning intracrystalline lattice defects gliding and climbing through the crystal lattice, which is known as dislocation creep (Shoji and Higashi, 1978; Schulson and Duval, 2009; Faria et al. 2014b). Ice Ih is the ice polymorph that occurs on Earth. It has a hexagonal symmetry and dislocation glide is primarily on planes perpendicular to the c-axes (i.e. basal planes) or on pyramidal or prismatic planes. Dislocation glide in ice Ih is characterized by a strong visco-plastic anisotropy, with resistance to glide on basal planes at least 60 times smaller than

on other planes (Duval et al., 1983). The strong preference for basal glide usually leads to an approximately single maximum crystallographic preferred orientation (CPO) with the c-axes mostly aligned with the direction of maximum finite shortening (Azuma and Higashi, 1985). Early experimental studies by Steinemann (1954) show, that such a single-maximum CPO causes a mechanical anisotropy of a deformed aggregate of ice grains. This is supported by experiments by Gao and Jacka (1987), the review by Budd and Jacka (1989) and more recent studies by Treverrow et al. (2012).

Visco-plastic deformation of ice is accompanied by recrystallisation (Duval, 1979; Jacka and Li, 1994; Faria et al. 2014b), as is common in minerals at high homologous temperatures. Recrystallisation processes have direct implications on creep behaviour as they affect fabric development and hence the flow of ice (Duval and Castelnau, 1995; Castelnau et al., 1996; Duval et al., 2000). The nomenclature to describe recrystallisation and microstructure varies between glaciology, geology and material science. For consistency, in this manuscript we employ the terminology proposed by Faria et al. (2014b).

Under static conditions and non-deformation related, *normal grain growth* or static grain boundary migration driven by surface energy minimisation (Stephenson, 1967; Gow, 1969; Duval, 1985) leads to a microstructure with only slightly curved grain boundaries and 120° angles at grain triple junctions (foam texture). The resulting grain size distribution is log-normal according to (Humphreys and Hatherly, 2004, pp. 334-335). On the contrary, *strain-induced boundary migration* (*SIBM*) as described by Duval et al., (1983) or Humphreys and Hatherly (2004, pp. 251-253) minimizes stored strain energy by migrating

boundaries towards less strained neighbouring grains. Intracrystalline annihilation of dislocations by lattice re-orientation into lower energy configurations, known as *recovery* (White, 1977; Urai et al., 1986; Borthwick et al., 2014), additionally lowers stored strain energies. Recovery accompanied by gradual formation of subgrain boundaries and ultimately new high angle grain boundaries (e.g. *polygonisation,* Alley et al., 1995) is termed *rotation recrystallisation* (Passchier and Trouw, 2005). These recrystallisation phenomena operate during deformation are summarized by the term *dynamic recrystallisation*.

Recrystallisation processes operate concurrently, but the proportion of contribution of each mechanism varies. The dynamic recrystallisation diagram by Faria et al. (2014b) describes the relative contributions as a function of strain rate and temperature, as was done before for quartz (Hirth and Tullis, 1992). According to these models, rotation recrystallisation is more dominant with higher strain rates whereas strain-induced boundary migration dominates at higher temperatures.

In very shallow firn, at mass densities below 550 kg m$^{-3}$, compaction by displacement, re-arrangement and shape-change of snow particles is attributed to grain boundary sliding (Alley, 1987), neck growth between grains by isothermal sintering (Blackford, 2007) and temperature gradient metamorphism (Riche et al., 2013). Once the critical density is exceeded, the dominating mechanism becomes plastic deformation by intracrystalline creep (Anderson and Benson, 1963; Faria et al., 2014b). For the EPICA Dronning Maud Land (EDML) ice core, this critical density is reached at around 20 m depth (Kipfstuhl et al., 2009). However, more recent tomographic analyses on EDML samples by Freitag et al. (2008) provide evidence for an early onset of plastic deformation at shallow depths of 10 m. Simulations by Theile et al. (2011) suggest an even shallower onset of plastic deformation and the absence of grain boundary sliding.

One way to determine which deformation mechanisms operate is to study the microstructure of the deformed material (Passchier and Trouw, 2005; Kipfstuhl et al., 2009; Faria et al., 2014a, b). Apart from experiments, numerical simulations are increasingly used as a tool to establish the link between deformation mechanisms, boundary conditions and resulting microstructures (see review of Montagnat et al., 2014). Unfortunately, most studies on ice deformation only considered pure ice without air bubbles. Some exceptions are the experimental studies of Arena et al. (1997) and Azuma et al. (2012), or the numerical simulations of Roessiger et al. (2014) on grain growth of ice in the presence of air inclusions. Recent numerical modelling by Cyprych et al. (2016) indicates the importance of strain localisation in polyphase materials, but does not include a description of microstructure evolution during recrystallisation. Systematic numerical studies of the effect of a second phase, in this case air, on plastic deformation and concurrent microstructure evolution during dynamic recrystallisation are still lacking.

In this contribution we investigate the implications of air inclusions on deformation and recrystallisation to assess the importance of dynamic recrystallisation at shallow levels of ice sheets. For that purpose, we for the first time employ an explicit numerical approach combining both polyphase crystal plasticity and recrystallisation. Particular focus is given to two microdynamical aspects, which are (1) the strain distribution in the polyphase and polycrystalline ice-air aggregate and (2) its relation to (deformation induced) dynamic recrystallisation.

## 2 Methods

### 2.1 Multi-process modelling with Elle

We used the open-source numerical modelling platform Elle (Bons et al., 2008; Jessell et al., 2001; Piazolo et al., 2010), as this code is very suitable to model the interaction of multiple processes that act on a microstructure. So far, Elle has been applied to a range of microdynamic processes, such as strain localisation and porphyroclast rotation (Griera et al., 2011; 2013), deformation of polyphase materials (Jessell et al., 2009) or folding (Llorens et al., 2013a, b). Recent applications of Elle codes utilized and updated for this study are on dynamic recrystallisation in pure ice (Llorens et al., 2016a, b), grain growth (Roessiger et al., 2011; 2014), and folding in ice sheets in relation to mechanical anisotropy, both on the small (Jansen et al., 2016) and large scale (Bons et al., 2016). To simulate visco-plastic deformation of the polyphase and polycrystalline aggregate

with concurrent recrystallisation, the full field crystal visco-plasticity code VPFFT by Lebensohn (2001) was coupled to implementations of recrystallisation processes in Elle using the approach described in Llorens et al. (2016a, b). Here we only briefly explain the essentials of the modelling technique. The reader is referred to Jessell et al. (2001) and Bons et al. (2008) for the general principles of Elle. Details of the algorithms for grain boundary migration can be found in Becker et al. (2008) and Roessiger et al. (2011; 2014), and for coupled VPFFT and recrystallisation in Llorens et al. (2016a, b).

## 2.2 Discretisation of the microstructure

The two-dimensional microstructure of ice and air inclusions is discretised in a contiguous set of polygons with fully wrapping and periodic boundaries (Fig. 1a; Llorens et al., 2016a.b; Bons et al., 2008). In the setup used here, the polygons (termed *flynns*) are either ice crystals or air inclusions. Island grains such as a grain inside another grain are not allowed in Elle, for topological reasons. *Flynns* are delimited by straight segments that join boundary nodes (*bnodes*) in either double- or triple-junctions. Quadruple or higher-order junctions are also not allowed in Elle. Additionally, we superimpose a regular grid of unconnected nodes (*unodes*) on the set of *flynns*. *Unodes* store local state variables such as stress, normalized von Mises strain rate or dislocation density. Crystallographic orientations at *unodes* are defined by Euler triplet angles, following the Bunge convention. After each deformation increment, all state variables are mapped back to a regular, rectangular *unode* grid, as this is required by the VPFFT code. To track the finite deformation, a second set of *unodes*, on an initially regular square grid, represent material points or passive markers that are displaced each deformation step.

Topology checks are carried out at all times during a simulation to ensure compliance with topology restrictions and to maintain the set resolution. These include keeping *bnode* distances between a minimum and maximum separation by either deleting or inserting *bnodes* (Fig. 1b), and removing *flynns* that are smaller than a set minimum area or contain no *unodes* (Fig. 1c). To avoid the formation of a quadruple junction, a neighbour switch is performed between triple junctions closer than the minimum separation distance (Fig. 1c). When two sides of a *flynn* approach each other to below a set minimum distance, the *flynn* is split into two (Fig. 1e). This allows bulging grain boundaries to sweep across entire grains without causing overlapping *flynns*.

## 2.3 Visco-plastic deformation using full field approach

The full field crystal visco-plasticity code (VPFFT) by Lebensohn (2001) was coupled to the Elle numerical modelling platform following the approach by Griera et al. (2013) and Llorens et al. (2016a). The approach is based on calculating the mechanical field (i.e. stress, strain rate) from a kinematically admissible velocity field that minimizes the average local work-rate under the compatibility and equilibrium constraints (Lebensohn 2001; Lebensohn et al., 2009; Griera et al. 2013).

In this approach, intracrystalline deformation is assumed to be accommodated by dislocation glide on pre-defined slip systems, using a non-linear viscous, rate-dependent law. The strain rate $\dot{\varepsilon}_{ij}(\boldsymbol{x})$ at each position $\boldsymbol{x}$ (*unode*-position) in the grid is essentially the sum of the shear strain rates on all $N$ slip systems (Eq. 1):

$$\dot{\varepsilon}_{ij}(\pmb{x}) = \sum_{s=1}^{N} m_{ij}^{s}(\pmb{x})\dot{\gamma}^{s}(\pmb{x}) = \dot{\gamma}_0 \sum_{s=1}^{N} m_{ij}^{s}(\pmb{x}) \left( \frac{m_{ij}^{s}(\pmb{x})\sigma_{ij}'(\pmb{x})}{\tau^{s}(\pmb{x})} \right)^3$$

(1)

The constitutive equation relates the shear rate $\dot{\gamma}^{s}$ on each slip system ($s$), relative to a reference shear rate $\dot{\gamma}_0$, to the deviatoric stress $\sigma_{ij}'$ and the orientation of the slip system that is defined by the symmetric Schmidt tensor $m_{ij}^{s}$ (the dyadic product of a

vector normal to the slip plane and slip direction). The effective viscosity or "ease of slip" of each slip system is defined by the slip-system dependent critical resolved shear stress $\tau^{s}$. Here we use a stress exponent of three, assuming Glen's law (Glen, 1958).

Since the strain rate and stress fields are initially unknown, an iterative scheme is implemented with a spectral solver using a Fast Fourier Transformation. The VPFFT code provides the full velocity field, which is integrated to the displacement field

for a small time step, assuming velocities remain constant. The displacement field is applied to the passive marker grid and to all *bnodes* to apply the deformation to the grains and air inclusions. Lattice orientations are updated and remapped onto the rectangular *unode* grid. Furthermore, geometrically necessary dislocation densities are calculated using the plastic strain gradient following Brinckmann et al. (2006) and assuming constant Burgers vectors for all slip systems.

### 2.4 Recrystallisation

### 2.4.1 Polyphase grain boundary migration

Polyphase grain boundary migration is modelled using a front-tracking approach, which is explained in detail by Becker et al. (2008) and Llorens et al. (2016a). Grain boundary migration is achieved by moving individual *bnodes*. In general, the movement $\Delta \pmb{x}$ of a *bnode* is calculated from its mobility $M$ and driving force $S$ over a small numerical time step $\Delta t$:

$\Delta x = SM(\,M_0, T(°C)\,)\Delta t$

(2)

where the mobility $M$ is a function of temperature $T$ and intrinsic mobility $M_0$ (Nasello et al. 2005). The intrinsic mobility $M_0$ varies for different phase boundaries. The driving force $S$ is calculated from the change in local free energy ($dE$) resulting from a change in position ($dx$) of the *bnode* under consideration. $dE$ is a function of the change in boundary length and, hence, total local grain-boundary surface energy (Becker et al. 2008) and the change in stored strain energy (Llorens et al. 2016). A *bnode*

is moved (using Eq. 2) in the direction of maximum free-energy reduction, which is determined from four small orthogonal trial displacements of that *bnode*. For the movement of ice-air boundaries, the stored strain energy was not taken into account. In a polyphase aggregate, the conservation of mass requirement influences boundary migration. In theory, any local movement of a boundary needs to conserve the cross sectional area of its host grain. This restriction would inhibit most ice-air interface movements, prohibiting any geometrical changes of air inclusions. Therefore, an additional energy term ($E_{area}$) is introduced

to counter-act that the surface energy ($E_{surf}$) would drive an ice-air boundary inwards and let air inclusions shrink (Roessiger

et al., 2014). For ice-air boundaries, the total local energy ($E_{total}$) at a given trial position $j$ only depends on the surface energy change and the relative area change resulting from a theoretical movement of the boundary node to this position:

$$E_{total}(j) = E_{surf}(j) + E_{area}(j)$$

(3)

$$E_{area}(j) = c\left(\frac{A(j) - A_0}{A_0}\right)^2$$

(4)

where $A_j$ is the area of the air inclusion when the *bnode* is at trial position $j$ and $A_0$ is the initial inclusion area. Decreasing $c$ essentially increases the accepted violation of the conservation of mass requirement, allowing a stronger change in cross sectional area. For the polyphase aggregate of ice and air, the pre-factor $c$ can be regarded as a compressibility factor:

Theoretically, surface energy drives the bubble surface inward, compressing the enclosed air and increasing the air pressure in the bubble. This pressure would counter-act the surface movement until an equilibrium between surface tension and inner bubble pressure is reached, leading to a stable bubble cross sectional area. The factor $c$ controls how quickly this equilibrium is reached.

To fulfil the conservation of mass requirement, any movement of the interface that is not mass conserving should be inhibited.

However, this would cause a locking of the ice-air interface and inhibit any changes in inclusion shape. Therefore, we allow movements that conserve the overall fraction of air, but allow for sufficient shape changes of the bubbles. Preparatory tests yielded $c = 0.1$ as a compromise achieving this equilibrium. This assumption will inhibit almost all porosity changes during the simulation, causing our approach to assume an incompressible air inclusion, which does not allow us to quantitatively compare the modelled inclusion shapes or sizes with natural samples that experienced compaction. This limitation is further

discussed in section 4.5.

We chose to use most input parameters from published literature to allow comparability of the results to previous modelling. Following the experimental results of Nasello et al. (2005), the intrinsic mobility $M_{ice-ice}$ of ice-ice boundaries was set to 0.023 $m^4$ $J^{-1}$ $s^{-1}$, which Nasello et al. (2005) determined for slow movement and is consistent with previous modelling by Llorens et al. (2016a,b). A slower movement is more suitable for our simulations as a higher mobility would cause numerical errors when

using the same time step. Furthermore, it mitigates the effect of large numerical grain sizes (section 4.5). The grain boundary mobility was determined as a function of temperature and intrinsic mobility according to Nasello et al. (2005). To be consistent with previous modelling, the surface energy $\gamma_{ice-ice}$ of ice-ice boundaries was set to 0.065 J $m^{-2}$, as commonly used in the literature (Ketcham and Hobbs, 1969; Nasello et al., 2005; Roessiger et al., 2014; Llorens et al. 2016a,b). Based on Roessiger et al. (2014), the mobility ratio of ice-ice and ice-air boundaries $M_{ice-ice}/M_{ice-air}$ was set to 10, which in their study provided

results in compliance with the experimentally derived grain growth rates of Arena et al. (1997). The surface energy $\gamma_{ice-air}$ for ice-air boundaries was set to 0.52 J $m^{-2}$, which as a function of ice-ice surface energies ($\gamma_{ice-ice}$) results in a dihedral angles of 173° and almost circular air inclusions (Roessiger et al., 2014).

In a two-phase model, such as ice with air inclusions, three boundary types are possible: Ice-ice, air-ice and air-air boundaries. Air-air boundaries can occur in the model, for example when two air inclusions merge into one. These boundaries are purely numerical and have no physical meaning. They are therefore excluded them from any modelling processes or post-processing analyses.

During the simulation, all *bnodes* are selected in a random order and moved according to Eq. (2) one at a time. After each movement, topological checks are performed in keeping with the topological restrictions of Elle and to avoid impossible topologies such as *bnodes* sweeping across other grain boundary segments. Once a *unode* is swept by a moving boundary and thus changes its host grain, its dislocation density is set to zero and its lattice orientation to the value of the nearest neighbour *unode* in the new host grain.

**2.4.2 Rotation recrystallisation**

The process of rotation recrystallisation is modelled in two separate steps during the multi-process simulation: (1) Recovery by rearranging the intracrystalline lattice orientations into lower energy configurations such as subgrain boundaries, which is the predecessor for (2) the creation of new grains defined by high angle boundaries, which here implies inserting new boundary nodes and splitting an existing *flynn*.

In analogy to the grain boundary migration code, an energy minimisation system is used. Each *unode* is regarded as a small crystallite characterized by a lattice misorientation with respect to its first-order neighbours. Misorientation is the difference in lattice orientation between two *unodes*, which increases the total free energy of the local system. Small trial rotations are used to determine which lattice rotation would result in the maximum decrease in local free energy. The lattice in the *unode* is then rotated according to this decrease and a "mobility" term, as described in detail in Borthwick et al. (2014) and Llorens et

al. (2016a).

Both the visco-plastic deformation and the above recovery process lead to polygonisation, i.e. the formation of new high-angle grain boundaries defined by a lattice misorientation between neighbouring *unodes* that exceeds a critical angle $\alpha_{hagb}$. Such new grain boundaries are initially not defined by *bnodes*, and are thus numerically excluded from grain-boundary migration (Section 2.4.1). Polygonisation requires the creation of new high angle grain boundaries by splitting an existing *flynn* and

25 inserting new boundary nodes. When intragranular misorientations that exceed $\alpha_{hagb}$ are detected, grain splitting is activated. This is achieved by finding clusters of *unodes* with common lattice orientations, separated by high-angle boundaries. The positions of the new boundary nodes are found using a Voronoi decomposition of the *unode* clusters, storing the Voronoi points surrounding the cluster as new *bnodes*. The critical angle $\alpha_{hagb}$ has been suggested to be 3 to 5° for ice Ih, based on experiments that combine grain boundary properties and high-angular resolution measurements (Weikusat et al., 2011a, b).

Here we use $\alpha_{hagb} = 5°$ as a conservative estimate. Lower angles would lead to smaller grain sizes, which potentially cause more topological problems.

## 2.5 Process coupling

Multi-process modelling of polyphase deformation and recrystallisation is achieved by operator splitting. In Elle, the specific physical processes that contribute to microstructure evolution are programmed as standalone modules. These are coupled by a control program, successively running them in isolation, each for a short numerical time step (Fig. 2a). The numerical setup takes into account the visco-plastic deformation (Section 2.3) and dynamic recrystallisation (DRX, Section 2.4) which here covers grain boundary migration, recovery and polygonisation.

The recrystallisation modules are computationally less expensive but require short numerical time steps. To reduce numerical errors, the time step for recrystallisation processes is set 20 times smaller than for deformation (VPFFT). In accordance with the smaller time step, one simulation step comprises one VPFFT step and five subloops that run the recrystallisation codes four times per subloop. This adds up to 20 times more recrystallisation steps than VPFFT steps, but an equal time step for all physical processes. Systematic studies showed that the order of the processes as illustrated in Fig. 2a has no significant influence on the results using the properties described above.

## 2.6 Setup of simulations

Three starting microstructures were used to investigate the effect of visco-plastic deformation and DRX at different area fractions of air inclusions of 0 %, 5 % and 20 % air phase, termed F00, F05 and F20, respectively (Fig. 2b). In general, we refrain from relating our air fractions to specific depth or porosity ranges in firn or ice as our approach is limited and assumes an incompressible air phase. We chose to use these settings to represent only approximate ice-air aggregates as found in firn (F20) and well below the firn-ice transition (F05) and used simulation F00 for reference. With this, we do not limit our study to firn, but to ice-air aggregate in general. All initial microstructures were created from the same 10x20 cm² foam texture with 3267 grains. With this, the intial mean grain sizes of our modelling were larger than typical firn mean grain sizes as presented in e.g. Kipfstuhl et al. (2009), but consistent with previous simulations by Llorens et al. (2016a,b). Lattice orientations were mapped onto a regular grid of 256x256 *unodes* with a random initial lattice orientation assigned to each grain. Air inclusions were introduced by setting air properties to the desired area percentage of grains, followed by running solely surface-energy based (static) polyphase grain boundary migration until air inclusion sizes equilibrated in area and shape. For consistency, for this static grain boundary migration, the same area energy factor $c = 0.1$ (Eq. (4)) as for the actual dynamic recrystallisation and deformation simulations was used. Section 4.5 discusses the use of this factor in more detail.

The three starting microstructures were deformed in pure shear with a constant incremental strain of 1% vertical shortening over 75 simulation steps. Each simulation step comprised 20 recrystallisation steps per VPFFT step and equalled $10^8$s = 3.16 yrs, resulting in a vertical strain rate of $10^{-10}$ s$^{-1}$ and deformation up to 53% vertical shortening. We remark that the modelled strain rate is about an order of magnitude faster than assumed for firn at the EDML site (Faria et al., 2014b) and that modelling a slower strain rate is possible, yet currently too numerically expensive. From a technical point of few, fast strain rates have

the advantage that the time steps for recrystallisation routines can be small. To achieve slower strain rates, the number of recrystallisation steps per deformation step would need to be increased at a significant expense of computation time.

Dislocation glide was assumed for ice Ih crystallography with slip on basal, pyramidal and prismatic planes, using a ratio of basal to non-basal critical resolved shear stresses of $\tau_{basal}/\tau_{non-basal} = 20$. Air was modelled as an incompressible crystalline material with the same crystallography and slip systems as for ice, but with $\tau_{s-air}$ set 5000 times smaller than $\tau_{basal}$ of ice. Hence, as for ice Ih, deformation in the air phase was also resolved on basal, pyramidal and prismatic planes which were characterized by equally small critical resolved shear stresses. This leaves the air phase slightly anisotropic as the deformation is restricted to these defined slip planes. However, this approximation does not significantly affect the results (section 4.5 and supplementary figure S2). With this treatment of the air phase, the stress in points belonging to pores can be considered to nearly vanish compared to stresses reached in the solid grains, which is coherent with results from modelling of void growth using a dilatational visco-plastic FFT-based formulation (Lebensohn et al, 2013). The simplification of assuming incompressibility allowed us to exclude the effect of compaction during microstructure evolution and is further discussed in section 4.5.

Temperature throughout the simulations was assumed constant at -30 °C. A detailed summary of all input properties can be found in Table 1. Where not indicated differently, we employed input parameters as used by Llorens et al. (2016a, Table 1). For comparison of grain size statistics, we additionally performed three normal grain growth (NGG) simulations using solely surface energy driven grain boundary migration and no deformation. The NGG simulations used the three microstructures for F00, F05 and F20 as presented in Fig. 2b and the numerical time step was kept the same as in the deformation simulations.

## 2.7 Post processing

### 2.7.1 Strain rate and strain localisation quantification

In order to visualize and explain the simulations, some post-processing steps were necessary. Strain rate tensor fields predicted by FFT were transformed in *von Mises equivalent strain rates* normalizing the von Mises strain rate for each *unode* to the bulk value of the whole model. The von Mises strain rate $\dot{\varepsilon}_{vM}$ provides a scalar measure of strain rate intensity and was calculated as a function of the symmetric strain rate tensor:

$$\dot{\varepsilon}_{vM} = \sqrt{\frac{2}{3}\dot{\varepsilon}_{ij}\dot{\varepsilon}_{ij}}$$

(5)

In addition, we quantified strain localisation at each step during the simulation. Analogous to Sornette et al. (1993) and Davy et al. (1995), the degree of localisation $F$ was calculated with:

$$F = 1 - \frac{1}{N_u}\frac{(\sum \dot{\varepsilon}_{vM})^2}{\sum \dot{\varepsilon}_{vM}^2}$$

(6)

where $N_u$ is the total number of *unodes* within ice grains and $\dot{\varepsilon}_{vM}$ the von Mises equivalent strain rate of each *unode*. The localisation factor $F$ ranges from 0 to 1, such that 0 represents completely homogeneous deformation and $1-1/N_u$ maximum localisation, where all strain is accommodated by a single *unode*. Note that Sornette et al. (1993) and Davy et al. (1995) used a slightly different localisation factor $f=1-F$, where 1 represents homogeneous deformation.

## 2.7.2 Driving forces and crystallographic orientations

For each step of grain boundary migration, the driving forces for migration were stored in *bnode* attributes differentiating between surface and stored strain energy driving forces. Details about the driving force calculation can be found in Llorens et al. (2016a, equations 10-12). By normalizing the local stored strain energy to *bnode* mean surface energy, we obtained a quantitative measure of how much grain boundary migration is induced by strain energy. For each simulation step, the *bnodes* only stored the driving forces for the last grain boundary migration step. This allowed to capture the correct driving forces at the end of the simulation step after a time increment during which strain energy was induced by deformation and reduced by recrystallisation processes. Hence, we determined a minimum estimate for strain energies, which may have been higher in an intermediate stage of the simulation step.

Crystallographic preferred orientations were stored and updated during the simulations. Pole figures and Eigenvalues were extracted using the texture analysis software MTEX (Bachmann et al., 2010; Mainprice et al., 2011) based on the orientation distribution function. The projection plane was chosen to be parallel to the x-y plane of the numerical model and c-axis orientation was expressed using the angles of azimuth and dip in this plane.

## 3 Results

Table 2 and Fig. 3 provide an overview of the results obtained from simulating pure shear deformation with ongoing recrystallisation for three different amounts of air inclusions. Selected movies illustrating the full microstructure evolution can be found as supplementary material in the AV Portal of TIB Hannover (av.tib.eu). The resulting microstructures are characterized by heterogeneous grain size distributions and a slight increase in average grain sizes compared to the initial one. Most grains have smoothly curved boundaries and are usually equidimensional to slightly elongate in the *x*-direction. Coalescence of air inclusions (Roessiger et al., 2014) led to a number of large inclusions in simulation F20. The largest air inclusions show a marked elongation, mostly oblique to the shortening direction. Small inclusions remained circular.

The strain and strain rate distribution is difficult to discern from the final air inclusion and grain shapes only, as these are constantly reworked by DRX (Llorens et al., 2016b). Instantaneous strain rate maps and finite strain passive marker grids provide better insight in deformation heterogeneity (Fig. 3), which is also visible in movies that show the whole deformation history (AV Portal of TIB Hannover, av.tib.eu). Strain localisation is observed in all simulations independent of the presence of air inclusions. While instantaneous strain rate maps (Fig. 3b) can only show localisation at the current time step, passive

marker grids reflect the accumulated strain throughout the microstructure evolution (Fig. 3a-b). Figure 3b shows that zones with high strain rates, are oriented at ≤45° to the shortening direction. Zones with a high finite strain, or shear bands, are visible in the finite-strain pattern. These zones of accumulated shear strain initially formed at ca. 45° and subsequently rotated away from the shortening direction, especially in air-free ice (F00). High strain (-rate) zones form bridges between air inclusions

when these are present. Regions between the high strain zones are characterized by both low strain rates and low accumulated finite strains.

Using the localisation factor $F$, it is possible to quantify the degree of strain localisation in our simulations. Figure 4 shows the evolution of this factor with strain for all simulations. In accordance with strain rate maps and passive marker grids, non-zero ($F≥0.3$) values are observed throughout the simulations, indicating strain localisation in all cases. $F$ increases up to about 40%

vertical shortening, after which the rate of increase is lower. Localisation increases with the amount of air inclusions, with $F≈0.5$-07 in simulation F20 about double that for pure ice (F≈0.3-0.35).

To investigate the competition between surface and stored strain energies in grain boundary migration, the strain energy driving forces for simulation F20 were normalized to mean surface energies and plotted for each *bnode* (Fig. 5a). The colour scale is adjusted to plot *bnodes* without a contribution of strain-induced energies in the background colour (blue). Boundaries with a

significant contribution of stored strain energies are indicated by green to red colours. A comparison with Fig. 3 shows that grain boundary migration that is driven mostly by strain energy (bright colours) are predominantly located in high strain (-rate) zones. Examples are indicated by large arrows in Fig. 5a and between three elongated and large air inclusions in the lower middle region of the final microstructure of F20. Conversely, the contributions of surface and strain energy to grain-boundary migration are about equal in less-strained areas.

Qualitatively, microstructure images show a heterogeneous grain size distribution (Fig. 3). To further visualize the spatial distribution of grain sizes, the microstructure in Fig. 5b shows ice grains coloured according to their area. Analogous to the driving force distribution for grain-boundary migration, we observe the smallest grains between air inclusions coinciding with zones of marked strain localisation.

Figure 6 depicts ice grain size statistics for the final microstructures. To visualise the influence of dynamic recrystallisation,

grain size histograms are compared with normal grain growth (NGG) simulation results. All simulations show an increase of average grain areas with respect to the initial mean grain sizes (Fig. 6, Table 1). However, dynamic recrystallisation simulations resulted in a grain size distribution skewed towards smaller grain sizes than for NGG simulations. Furthermore, the distribution of grain sizes is broadened when dynamic recrystallisation was active. With increasing amount of air in simulations F05 and F20, the average grain-size increase compared to the initial state is lower for both dynamic recrystallisation and NGG results.

Crystallographic preferred orientations in simulations are visualised using pole figures and maps of c-axis azimuths (Fig. 7). The evolution of orientations is also illustrated in movies to be found in the AV Portal of TIB Hannover (av.tib.eu). After 53% of vertical shortening, the initially random fabric is re-arranged with c-axes preferentially oriented parallel to the vertical shortening direction (Fig. 7a). This maximum becomes less pronounced with increasing amount of air as reflected in pole figures and quantified by a decrease in first eigenvalues of the orientation distribution from 0.80 (F00) to 0.69 (F20). This

trend is also visible in c-axis orientation maps (Fig. 7b), which shows a more heterogeneous distribution of well-aligned and random fabrics with increasing air content. Most grains in simulation F00 have c-axes azimuths parallel to the y-axis. Grains within high strain bands by a slight tilt of the c-axes to the left or right, depending on the orientation of the shear bands. In contrast, simulation F05 and, even more, F20 show areas of small grains with c-axes strongly aligned perpendicular to high

strain bands (white arrows in Fig. 7b). This means that the basal planes are aligned parallel to these bands. In low strain areas, such as the middle part of the F20 model, a much more random c-axis distribution is observed (white circle in Fig. 7b) compared to simulation F00.

## 4 Discussion

### 4.1 Strain localisation

Our simulations indicate a distinct strain localisation in both pure ice and ice with bubbles. Strain localisation is not a transient effect, but actually increases, at least up to about 40% of strain (Fig. 4). Strain localisation in pure ice (F00) is related to the plastic anisotropy of the ice crystal. Grains, or clusters of grains, with initially suitable orientations for slip accommodate strain more efficiently and thus initiate the first regions of strain-rate localisation. With progressive strain these localisation zones may strengthen as the basal planes align themselves with the local shear plane, or they are deactivated when either the internal

lattice orientations or the orientation of the localisation zones become less suitable for further localisation. Once deactivated, the localised zones only rotate and move passively with the bulk deformation, and may remain visible as shear bands in the finite strain grid. Our observation of strain localisation in a polyphase aggregate is consistent with numerical models by Cyprych et al. (2016), who predict strain localisation as an important mechanism in polyphase materials, such as ice with soft or hard inclusions.

In the presence of air inclusions, localisation zones are forming at bridges between the inclusions where stresses are highest. Even in the absence of plastic anisotropy this leads to the formation of localisation zones, especially in power-law materials (Jessell et al., 2009). With increasing air fraction, the arrangement and geometry of air inclusions become the main controllers of strain localisation in the ice-air aggregates and crystallographic orientations exert only a secondary control. The additional localisation mechanism causes stronger localisation in ice with air than without air.

The localisation zones enclose lozenge-shaped areas of low strain rate, which we term *microlithons*, in keeping with terminology used in geology (e.g. Passchier and Trouw, 2005 p. 78). In pure ice, the CPO within the microlithons is strong and trends towards a single maximum fabric. With air inclusions causing intensified strain localisation, the CPO is expected to be more heterogeneous with differences between high- and low-strain rate zones. Within localisation zones, the basal planes rotate towards the local shear plane, causing a divergence of the c-axes azimuths away from the vertical compression direction.

This is in contrast to the microlithons, in which CPO development is slower because of the relatively low strain rate. The weaker bulk single-maximum fabric with increasing amount of air is thus an effect of air inclusions causing distinctly localised zones that accommodate most of the deformation, and less deformed microlithons that preserve the initial fabric. Our results

are an illustrative example of the role of second phases on CPO development. If the weak phase is the secondary phase and the strong phase is load bearing, as in our simulations, strain localisation controlled by the distribution of second-phase inclusions. This produces a locally weaker CPO in the microlithons, but also in the bulk material.

Grain boundary sliding is assumed to be an explanation for a weaker CPO in polyphase materials (e.g. Fliervoet et al., 1997).
However, a weaker CPO with increasing content of the second phase is found in our simulations, in the complete absence of grain-boundary sliding. Therefore, a weaker CPO alone should not be regarded as unambiguous evidence for grain-boundary sliding. This supports studies indicating a very shallow onset of plastic deformation in the ice sheets (Freitag et al., 2008) and absence of grain boundary sliding (Theile et al., 2011).

## 4.2 Natural firn microstructures and numerical simulations

For comparison with natural ice and firn core samples, a microstructure image from 80 m depth from the EDML site is used (Fig. 8). For this image, a 2D sample was vertically cut from the firn core, and processed for microstructure mapping as described in Kipfstuhl et al. (2006; 2009) using a large area scan macroscope (Krischke et al., 2015). Here, we refrain from any detailed or quantitative comparison of our numerical simulations with respect to grain sizes, bubble size, shape or distribution. Such comparisons are hindered by model assumptions such as assuming an incompressible air phase or area
conserving pure shear deformation (section 4.5). We observed air inclusions coalescing during our simulations. This is expected from static polyphase grain-growth simulations by Roessiger et al. (2014), but may be suppressed in firn as simultaneous shrinkage of bubbles may hinder them from touching and merging.

The F20 and EDML microstructure show qualitative similarities (Fig. 8), in particular in the heterogeneity in grain shape and relative grain size distribution. As an estimate for numerical subgrain boundary density, average misorientation between
20 unodes was plotted together with the grain boundary network. In case of natural firn, detail area A (Fig. 8) contains larger grains, with more 120° angles at triple junctions and a lower density of subgrain boundaries than area B, which is characterized by a higher density of bubbles. Visually, similar areas can be found in simulation F20. Area C has large grains with straight grain boundaries and 120° angles at triple junctions and lower misorientations (qualitatively comparable to A), and area D is characterised by small grains relative to C. Triple junction angles in D differ from 120° and D has higher internal
misorientations (qualitatively comparable to B). The grain sizes can only be compared relatively, as mean grain size in simulation and sample differ. Besides, this natural sample is a 2D slice through a 3D body, but the simulations are purely 2D. For more rigorous comparisons, corrections for stereologic issues are required.

With respect to the limitations of the modelling approach (section 4.5), we cannot quantitatively compare the simulations to natural firn. However, as detail D (Fig. 8) is marked by higher finite strains related to strain localisation and is indicates more
influence of dynamic recrystallisation as presented by Kipfstuhl et al. (2009), similar processes may have controlled the natural microstructure in detail image B (Fig. 8). On the contrary, the natural ice in detail A probably experienced lower finite strains, as suggested from a comparison with the numerically modelled detail image C (Fig. 8). As strain localisation and the resulting heterogeneity in finite deformation pattern can be masked by grain boundary migration in natural ice (Llorens et al. 2016b),

our numerical simulations can help to visualise the actual heterogeneity within the structure, leading to an improved understanding of how dynamic recrystallisation is distributed within the ice-air aggregate.

## 4.3 Implications of strain localisation for the occurrence of dynamic recrystallisation

According to published deformation mechanism maps (Shoji and Higashi, 1978; Goldsby, 2006), dislocation creep is to be expected for the average grain size, strain rate and temperature of our simulations. Also, the effective density of the simulation F20 (approximately 750 kg m$^{-3}$) is above the critical density of 550 kg m$^{-3}$ where plastic deformation via dislocation creep is classically supposed to dominate (Maeno and Ebinuma, 1983). Therefore, we assume that our model assumption of deformation accommodated by dislocation creep only is sufficient to draw conclusions on mechanisms acting at comparable densities and depth in nature.

With the assumption of dislocation glide as the only strain accommodating mechanism, the dislocation density is expected to increase unless recovery reduces densities by re-arrangement of misorientations in lower energy configurations. A localisation in strain results in higher strain gradients at the localisation zone margins and hence locally higher strain energies. It is therefore associated with locally enhanced strain-induced boundary migration, as can be seen in Fig. 5a. This is in accordance with Weikusat et al. (2009) stating that strain-induced boundary migration occurs localized and the driving forces have to be considered locally. Duval (1985) argued that the strain energy in firn should be small in comparison with surface energies and Duval and Castelnau (1995) conclude that strain induced boundary migration is most dominant for temperatures of -10 °C or higher. This led to the assumption by De la Chapelle et al. (1998) that dynamic recrystallisation is essentially restricted to the basal part of ice sheets and therefore an improbable process in firn.

Faria et al. (2014b, p. 45) theoretically discuss the relation of strain localisation to localized dynamic recrystallisation in firn. They state that although the overall stresses and strains in firn are low, it is "characterised by large strain variability" and locally highly increased stresses and strains depending on the geometry of the air bubble network. They further conclude that stored strain energy could be very high in *particular* regions of the ice skeleton causing dynamic recrystallisation to start in shallow levels. Our simulations are coherent with this statement and confirm the theoretical predictions by Faria et al. (2014b). Since improving microstructural imaging methods by Kipfstuhl et al. (2006) gave further insight in firn microstructures, studies by Weikusat et al. (2009) and Kipfstuhl et al. (2009) gave microstructural evidence for dynamic recrystallisation in shallow parts of the ice column and firn. In contrast to assumptions by for instance De la Chapelle et al. (1998), it therefore seems probable that dynamic recrystallisation already takes place at very shallow levels in the ice sheet, at least in localised zones. Our simulations at -30°C as well as observations of natural firn microstructures at EDML ice coring site (Kipfstuhl et al. 2009) with approximately -45°C annual mean temperature (Oerter et al. 2009), indicate that even at low temperatures deformation is providing enough energy to allow for strain induced grain boundary migration. The relative dominance of this process is also a function of strain rate, since locally high strain rates and stress concentrations at bridges between the air inclusions induce high driving forces. This is in accordance with the recrystallisation diagram by Faria et al. (2014b), in which the occurrence

of recrystallisation mechanisms is essentially a function of temperature and strain rate (i.e. work rate, which is the product of stress and strain rate), rather than depth.

According to the dynamic recrystallisation diagram by Faria et al. (2014b), a lower strain rate would decrease the contribution of rotation recrystallisation and increase that of strain-induced grain boundary migration to the final microstructure. This would reduce the difference in grain size between high- and low-strain zones. A difference variation in grain size is, however, still observable in the EDML sample, indicating that grain-boundary migration was not able to obliterate the effects of rotation recrystallisation in these suspected high-strain zones, even at the lower natural strain rate.

Because of dynamic recrystallisation, grain shapes are mostly equidimensional, even in the highest-strain bands. Recrystallisation thus masks the localisation in the microstructure (Llorens et al., 2016b), making it difficult to discern strain localisation in natural samples. Subtle indications of localisation may, however, be zones with a deviating lattice orientation (Fig. 7) (Jansen et al., 2016) or zones with a smaller grain size (Fig. 5b). In single phase ice, where localisation zones shift through the material, only the youngest localisation zones may be visible, as the microstructure is reset in extinct localisation bands (compare simulation F00 in Fig. 3 and 7) (Jansen et al., 2016). Since bubbles fix the locations of shear localisation, their presence may be more obvious in natural samples, such as the one from the EDML (Fig. 8). In general, bubbles may only be useful to discern localisation zones (as in F20), if the bubbles are large and strain rates are high enough, which will cause elongated bubble shapes.

## 4.4 Grain size analyses

We refrain from a detailed comparison of our grain-size data and those observed in nature as a discussion of the stereologic issues related to our 2D model and sections through 3D natural samples would be beyond the scope of this paper. Furthermore, the limitations of the current modelling approach such as the large initial grain size, requirement of fast strain rates and incompressibility of air inclusions does not allow for such a detailed quantitative study. Still, the grain size statistics of the simulation results provide a comparison between the distribution for the non-deformation related normal grain growth (NGG) and deformation induced dynamic recrystallisation with varying amounts of air.

An increase in grain size is observed in all simulations, but less for dynamic recrystallisation and also less with an increasing amount of air inclusions. The observation of a lower final grain sizes for higher amounts of air is related to the growth regimes presented by Roessiger et al. (2014) based on numerical simulations on NGG in ice-air aggregates: (1) The first regime is characterised by ice grain sizes less than bubble spacing, where most grains can grow unhindered by bubbles as in single phase polycrystalline ice. The growth rate is constant. (2) The second regime is a transitional regime, where bubble spacing is close to the grain diameter and the growth rate decreases. (3) In the third regime, all grains are in contact with bubbles. A slow, but steady growth rate is reached again, controlled by the coalescence rate of bubbles that increases their spacing. In our case, NGG in the simulations with air inclusions is slowed down, indicating regime (2) growth with a small, but significant fraction of the grains in contact with inclusions, and thus hindered in their growth (Fig. 6).

In comparison with NGG simulations, our VPFFT simulations with dynamic recrystallisation show smaller final grain sizes and broader distributions. The broadening reflects the microstructural heterogeneity induced by dynamic recrystallisation (in particular grain splitting during rotation recrystallisation) and strain localisation. Locally, grains size has remained small due to rotation recrystallisation, whereas in other, low strain-rate regions, grain sizes have increased. Here surface energy

constituted a significant, if not dominant, proportion of the driving force for grain boundary migration. These results are consistent with the observed broadening of the grain size distribution with depth in firn from the EDML site (Kipfstuhl et al., 2009), accompanied by an increasing number of deformation-related substructures such as subgrain boundaries and irregular boundary shapes. Our modelling confirms the interpretation by Kipfstuhl et al. (2009) that this trend is related to the onset of dynamic recrystallisation.

**4.5 Limitations of the modelling approach**

In our polyphase simulations, air inclusions are modelled as an incompressible material. By imposing pure shear, we assume a deformation mode that conserves the total area and in turn the mass of both phases. However, in natural firn, most of the vertical thinning is achieved compaction. Compaction is a function of the surface energy driving movement of the ice-air interface and the counter-acting inner bubble pressure that depends on overburden pressure and bubble size and shape. Our

model assumes equilibrium between those pressures leading to a stable fraction of air. This is controlled by the area energy that is incorporated in the pre-factor $c$. The lower this factor, the more influence of surface energy is allowed and the more the conservation of mass requirement is violated as more inward movement of the ice-air interface is allowed. The pre-factor constant $c$ (Eq. 4) was adjusted to allow slight changes in cross sectional area that keep the overall amount of air in the model constant, but compensate shape changes due to deformation to maintain an approximately circular bubble shape and allowing

bubbles to merge. Preparatory tests yielded $c = 0.1$ as a suitable value to achieve this compromise. More detailed research is necessary to study the effect of a varying $c$ or scale it to natural ice.

We refrain from any study of depth evolution of porosity, inclusion shape or distribution. In fact, the numerical microstructure evolution cannot be regarded as an evolution with depth like in natural firn and ice. However, the model can be used to study the behaviour of firn or bubbly ice, independent of the history that led to the particular microstructure. At all times during the

simulation, we observe strain localisation controlled by air inclusions. This is even observed for small accumulated strains at low air contents (F05, Fig. 4). In addition, trial simulations showed that localisation also occurs at very different distributions of air inclusions at the same air fractions (see supplementary figure S1). There is no reason to expect that the strain localisation and elevated strain energies that drive dynamic recrystallisation that we observe for area-conservative pure shear would not occur during compaction.

Another approximation in the VPFFT approach is the treatment of the air phase. In the current model, air is treated as an ice Ih-symmetry crystal with equal basal, pyramidal and prismatic critical resolved shear stresses that are all 5000 times lower than for ice basal slip. This leaves the air phase slightly anisotropic. However, this assumption does not significantly affect the results since the effective contrast in slip resistance is significantly higher with a stress exponent of three (Eq. (1)). To further

investigate any effects on the results, we compared our approach with an updated VPFFT code that avoids any crystallography in the air phase by imposing zero stiffness to air *unodes*, thus causing their stresses to vanish (Lebensohn et al., 2011; 2013). This was done by applying both our and the updated VPFFT approach to the initial setup of F20 for an increment of 1% vertical shortening to compute the instantaneous strain rate and stress distributions. The results are essentially the same for both setups

(see supplementary figure S2) and show that the predictions of our simulations are not significantly affected by how we treat the air phase. Future simulations should include the optimized VPFFT approach imposing zero stiffness to air unodes.

Initial and final numerical grain sizes are larger than in natural firn, in particular with respect to the relatively high numerical strain rate. We chose to use initial grain sizes comparable to previous numerical simulations by Llorens et al. (2016a,b) and not to natural firn as other model assumptions would still hinder quantitative comparisons to natural samples. Adopting

different grain boundary mobilities can significantly alter the resulting grain size. Therefore, we chose to use accepted literature values for grain boundary mobility as experimentally derived by Nasello et al. (2005) and used for previous numerical simulations by Roessiger et al. (2014), Llorens et al. (2016a,b) and Jansen et al. (2016). Although the use of lower mobilities would decrease the predicted grain sizes, the use of accepted literature values is more justified with respect to the scope of the study. Atomistic processes driving recrystallisation may be decelerated in nature due to the presence of impurities and pinning

microparticles, which our simulation approach does not take into account. Any future comparison of simulations with natural ice may necessitate unexpectedly low or high values for material parameters such as an adapted lower grain boundary mobility, to achieve a more realistic grain size. Investigating a more suitable numerical mobility remains part of future developments. While the model scale affects the Elle recrystallisation processes and in turn grain sizes, the VPFFT approach is dimensionless and scale independent. The strain localisation bands and associated balance of recrystallisation driving forces is predicted by

the VPFFT routine. This implies, that the main observations and interpretations drawn in this paper in relation to strain localisation remain valid independently of the numerical grain sizes.

## 5 Conclusions

We used polyphase numerical models of deformation and recrystallisation to investigate the occurrence of dynamic recrystallisation in an air-ice composite such as polar ice and firn. To our knowledge this provides the first full-field numerical

simulation results on dynamic recrystallisation in polyphase crystalline aggregates in glaciology. We show that strain and strain-rate localisation is to be expected during ice deformation, forming shear bands that accommodate significant amounts of strain. Dynamic recrystallisation can occur at relatively shallow levels of the ice sheet where it is related to strain localisation and stress concentrations between the air inclusions. This results in an increased heterogeneity in ice sheet deformation and more dynamic recrystallisation activity than previously assumed. In fact, strain localisation is probably not the exception, but

the rule in ice sheets and glaciers. Wherever present, second phases such as air bubbles provide an effective mechanism for strain localisation in addition to mechanical anisotropy. Due to strain localisation, the rate of fabric change can be high locally, which is of special importance in firn, where bubbles are most abundant. The effects of localisation and heterogeneity in

distribution of firn recrystallisation and deformation could be considered in future firn densification models. Furthermore, as the utilized VPFFT approach is dimensionless, future research could investigate the probably large range of scales at which strain localisation may occur in glaciers and ice sheets.

**Acknowledgements**

We are thankful for support and helpful discussions with the members of the Elle community. We thank Till Sachau, Sepp Kipfstuhl and Johannes Freitag for their input to improve the manuscript as well as the helpful comments by two anonymous reviewers. This study was funded by the DFG (SPP 1158) grant BO 1776/12-1. Furthermore, we acknowledge funding by the Helmholtz Junior Research group "The effect of deformation mechanisms for ice sheet dynamics" (VH-NG-802) and traveling funds for presenting and improving this study by the EPICA Descartes travel price, the ESF research networking programme

on the microdynamics of ice (MicroDICE) and the Helmholtz Graduate School for Polar and Marine Research (POLMAR).

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

**Table 1: Input properties for the simulations F00, F05 and F20. Remaining input properties according to Llorens et al. (2016, Table 1). A more detailed description of the parameters is provided in sections 2.4 and 2.6 and Llorens et al. (2016a,b).**

| Symbol | Explanation | Input value |
|---|---|---|
| | Minimum *bnode* separation | $2.5 \times 10^{-4}$ m |
| | Maximum *bnode* separation | $5.5 \times 10^{-4}$ m |
| | Time step per simulation step | $10^8$ s |
| | Ratio of time step between VPFFT and recrystallisation codes | 20 |
| | Number of recrystallisation subloops per one step of VPFFT within one simulation step | 20 |
| | Incremental strain per simulation step | 0.01 |
| $\tau_{basal}\,/\,\tau_{non\text{-}basal}$ | Ice Ih: Ratio non-basal / basal glide resistance | 20 |
| $\tau_{basal}\,/\,\tau_{s\text{-}air}$ | Air: Ratio ice Ih basal resistance/air flow stress | 5000 |
| $M_{ice\text{-}ice}$ | Intrinsic mobility of ice-ice boundaries (Nasello et al., 2005) | 0.023 $m^4$ $J^{-1}$ $s^{-1}$ |
| $M_{ice\text{-}air}$ | Intrinsic mobility of ice-air boundaries (Roessiger et al., 2014) | 0.0023 $m^4$ $J^{-1}$ $s^{-1}$ |
| $\gamma_{ice\text{-}ice}$ | Ice-ice interface surface energy (Ketcham and Hobbs, 1969) | 0.065 J $m^{-2}$ |
| $\gamma_{ice\text{-}air}$ | Ice-air interface surface energy (Roessiger et al., 2014) | 0.52 J $m^{-2}$ |
| | Resulting dihedral angle at ice air triple junctions (Roessiger et al., 2014) | 173 ° |
| $\alpha_{hagb}$ | Critical misorientation: ice high angle boundary (Weikusat et al., 2010; 2011) | 5 ° |
| $c$ | Area energy or compressibility factor (10 times the value of Roessiger et al., 2014) | 0.1 |

**Table 2: Overview on numerical simulations using crystal visco-plasticity and dynamic recrystallisation (DRX) and only normal grain growth (NGG) simulations. NGG simulations used the same initial microstructures than DRX simulations.**

| | Area fraction of air | Initial number of ice grains | Final number of ice grains DRX (*NGG*) | Initial mean ice grain area and (*standard deviation*) | DRX: Final mean ice grain area and (*standard deviation*) | NGG: Final mean ice grain area and (*standard deviation*) | Final and (*initial*) first eigenvalue of CPO |
|---|---|---|---|---|---|---|---|
| F00 | 0 % | 3267 | 1631 (*1093*) | 6.12 mm² (*3.50 mm²*) | 12.17 mm² (*12.26 mm²*) | 18.30 mm² (*13.25 mm²*) | 0.7603 (*0.3393*) |
| F05 | 5 % | 3128 | 1994 (*1155*) | 6.07 mm² (*3.43 mm²*) | 9.44 mm² (*10.87 mm²*) | 16.43 mm² (*12.34 mm²*) | 0.6975 (*0.3390*) |
| F20 | 20 % | 2654 | 1891 (*1265*) | 5.96 mm² (*3.33 mm²*) | 8.30 mm² (*9.80 mm²*) | 12.53 mm² (*10.52 mm²*) | 0.5665 (*0.3468*) |

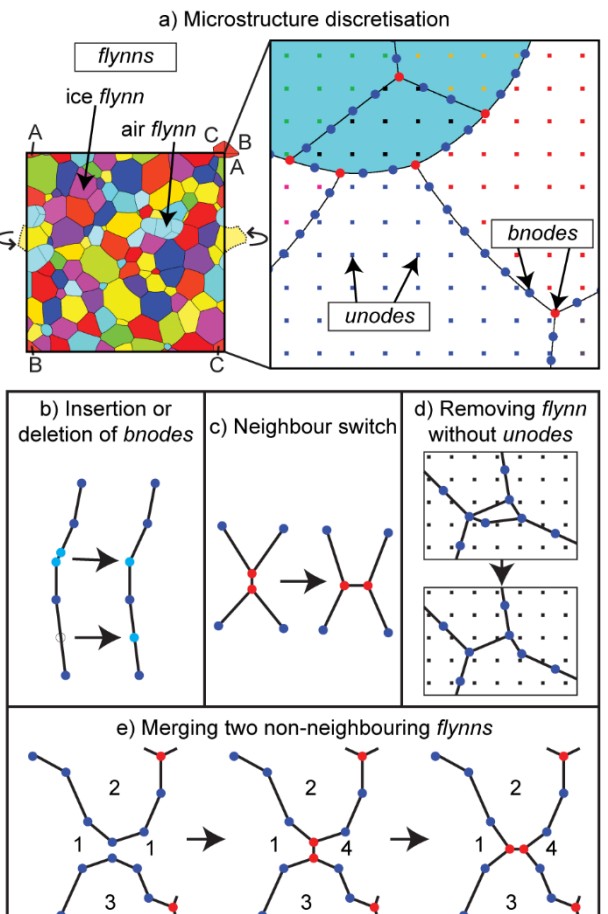

**Figure 1: Microstructure discretisation in Elle. (a) A contiguous set of polygons (flynns) is composed of boundary nodes (bnodes) and has periodic boundaries. An additional grid of unconnected nodes (unodes) is superimposed on flynns and bnodes to store intracrystalline properties, state variables and track deformation. (b-e) Topological checks performed to keeping topological restrictions in Elle. Checks (b), (c) and (e) are based on minimum and maximum bnode separations. Check (d) removes extremely small flynns that contain no undoes or have areas smaller than the area enclosed by four neighbouring unodes by merging them to a neighbour of the same phase.**

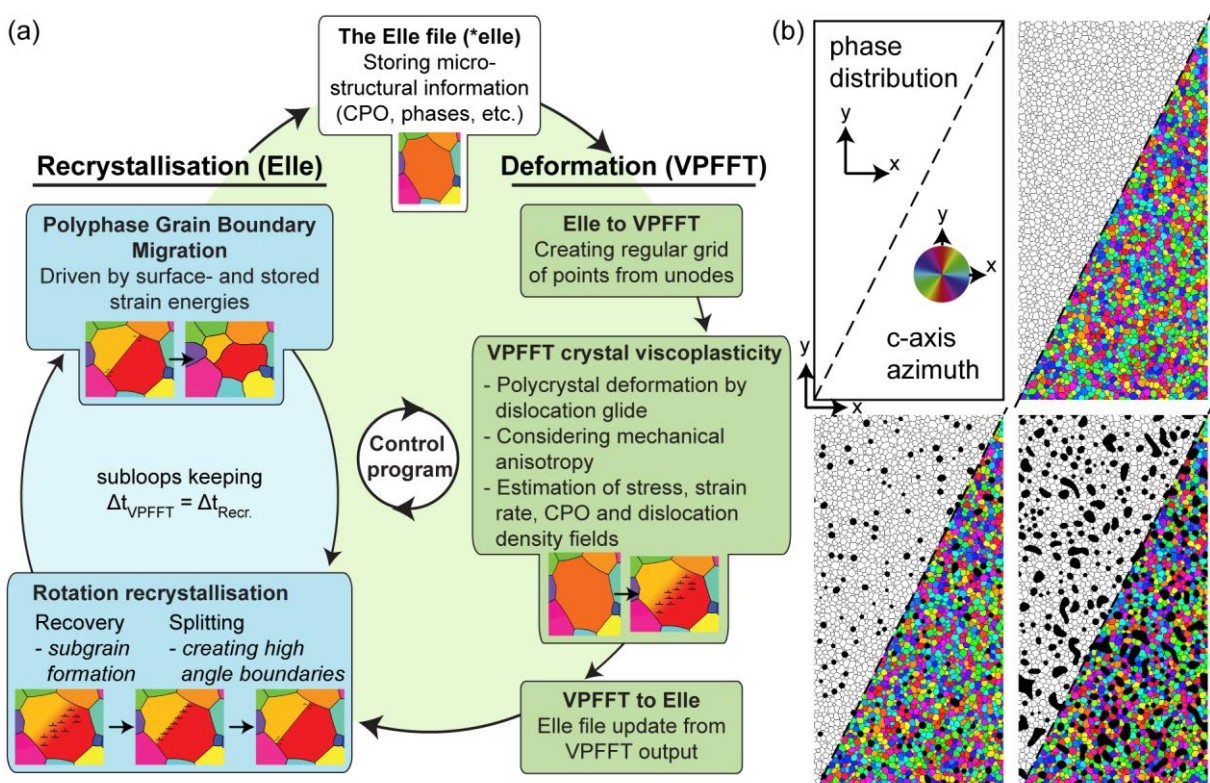

**Figure 2: (a) Multi-process modelling by operator splitting is achieved by successively running individual process modules. One step of deformation by VPFFT code is followed by five subloops (with shorter time steps) of recrystallisation, each comprising four steps of recovery and grain-boundary migration, to keep a constant time step for all combined processes. (b) Initial 10x20 cm microstructures with foam textures containing 0 %, 5 % and 20 % of air and with an initially random crystallographic fabric. Upper left half shows grain boundary network, lower right the lattice orientations.**

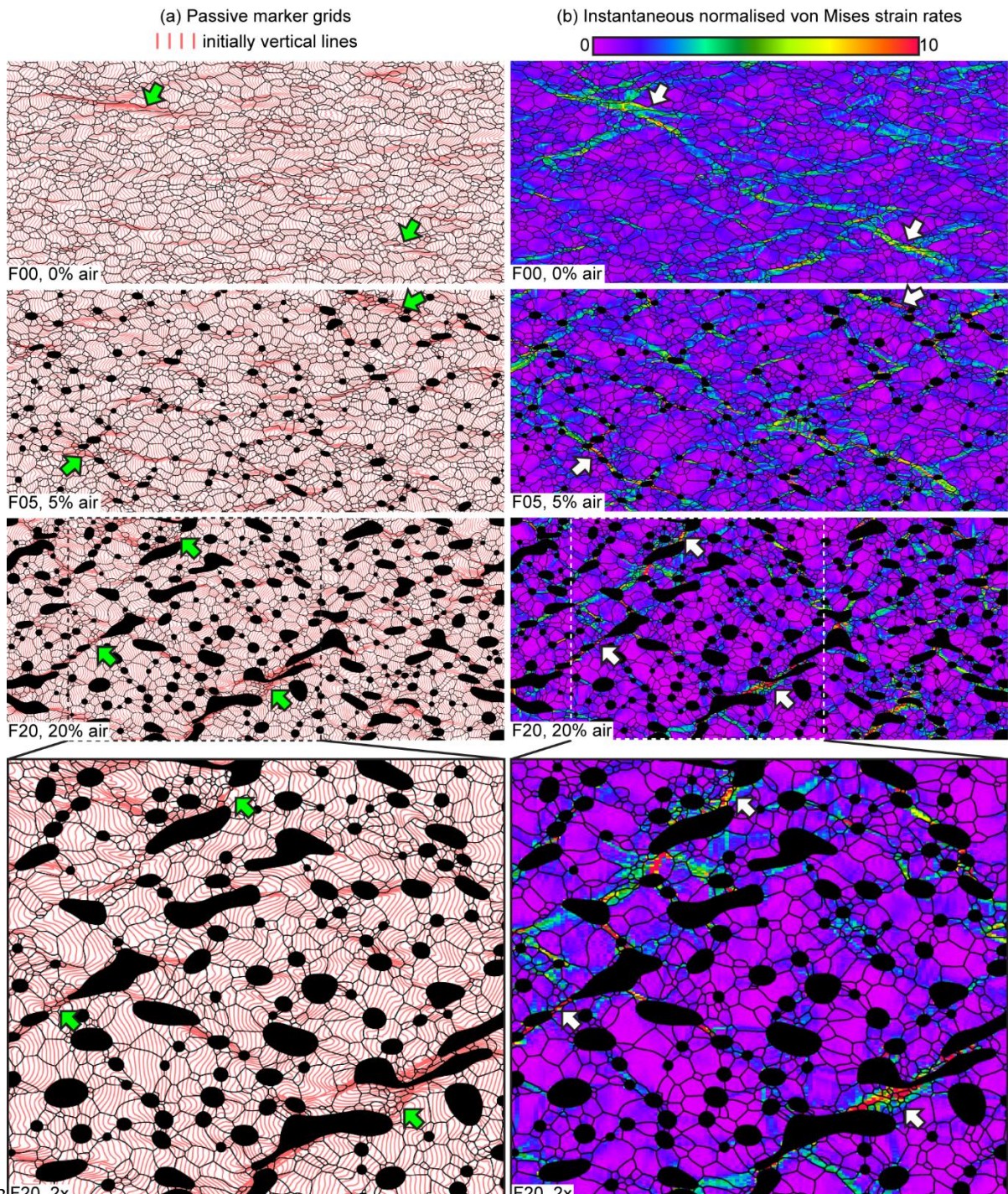

**Figure 3: Overview of modelling results at last time step at 53 % vertical shortening under pure shear conditions. (a) Grain boundary network superimposed on passive marker grid of initially vertical parallel lines to show the finite strain distribution. (b) The same microstructures superimposed on the map of instantaneous strain rates expressed as von Mises strain rates normalized to the bulk value. Arrows in both images indicate zones of marked strain localisation. Air inclusions are displayed in black.**

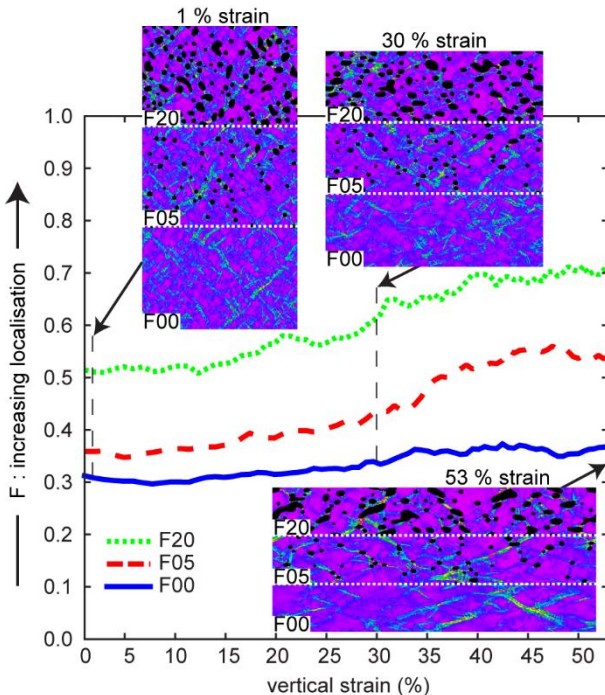

**Figure 4: Evolution of localisation factors (F) with strain quantifying strain localisation in the microstructures for all simulations. A factor of 0 represents homogeneous deformation, the factor increases towards one with increased strain-rate heterogeneity and localisation. The normalized von Mises strain rate maps at 1 %, 30 % and 53 % vertical strain are shown for reference. The maps are subdivided to show results of simulation F20 in the upper third, of F05 in the middle and F00 in the lower third part of the model box. They illustrate strain localisation at different stages of the simulation.**

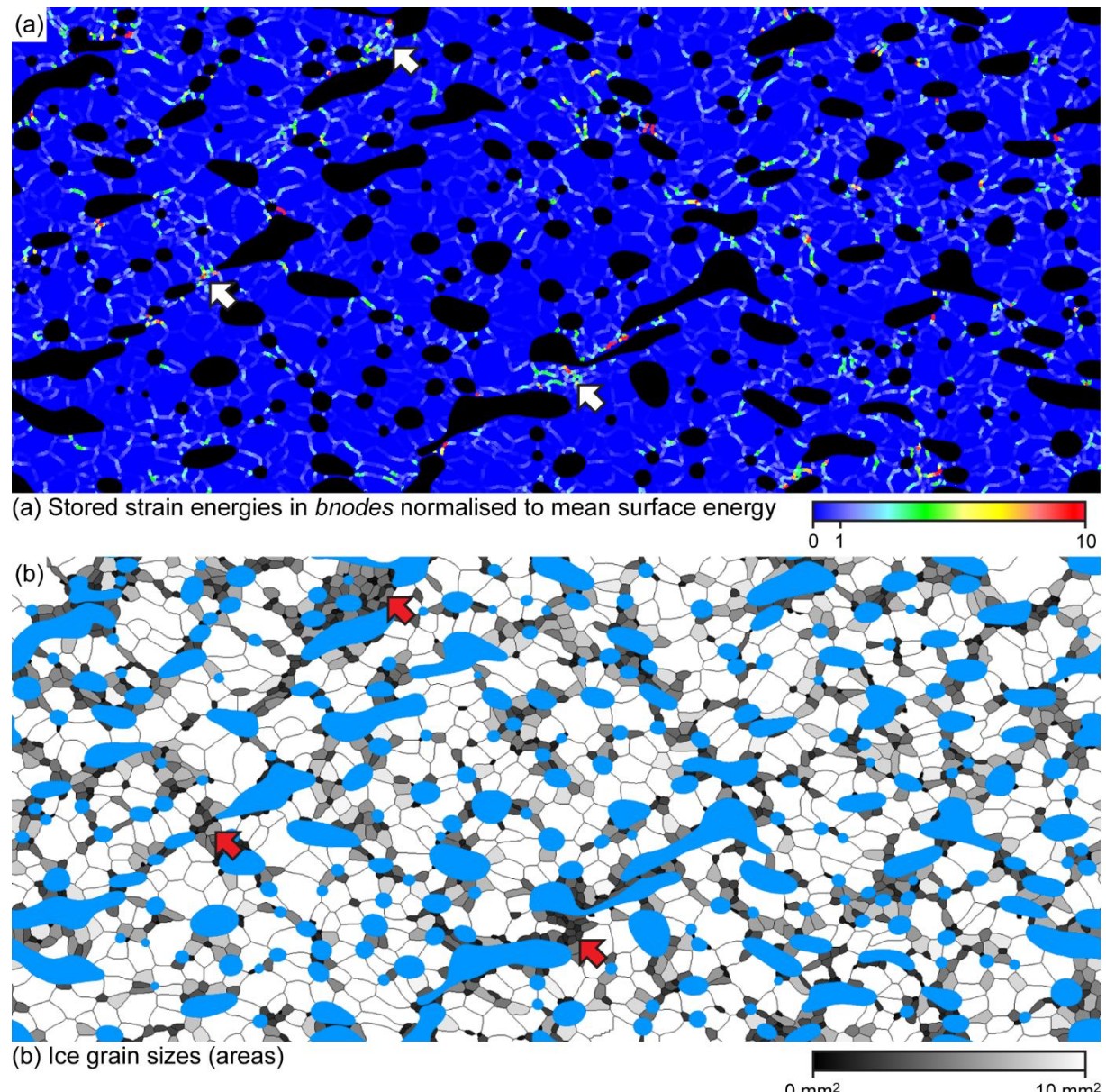

(a) Stored strain energies in *bnodes* normalised to mean surface energy

0  1                                    10

(b) Ice grain sizes (areas)

0 mm²                                    10 mm²

**Figure 5: Details of results of simulation F20 at 53% vertical shortening: (a) Boundary nodes colour coded according to the proportion of strain-induced boundary migration. Air inclusions are plotted in black since they do not contribute to strain induced boundary migration. (b) Microstructure colour-coded according to the areas of ice grains, with air inclusions displayed in blue. Smallest grains appear grey to black. Arrows in both images indicate zones of marked strain localisation.**

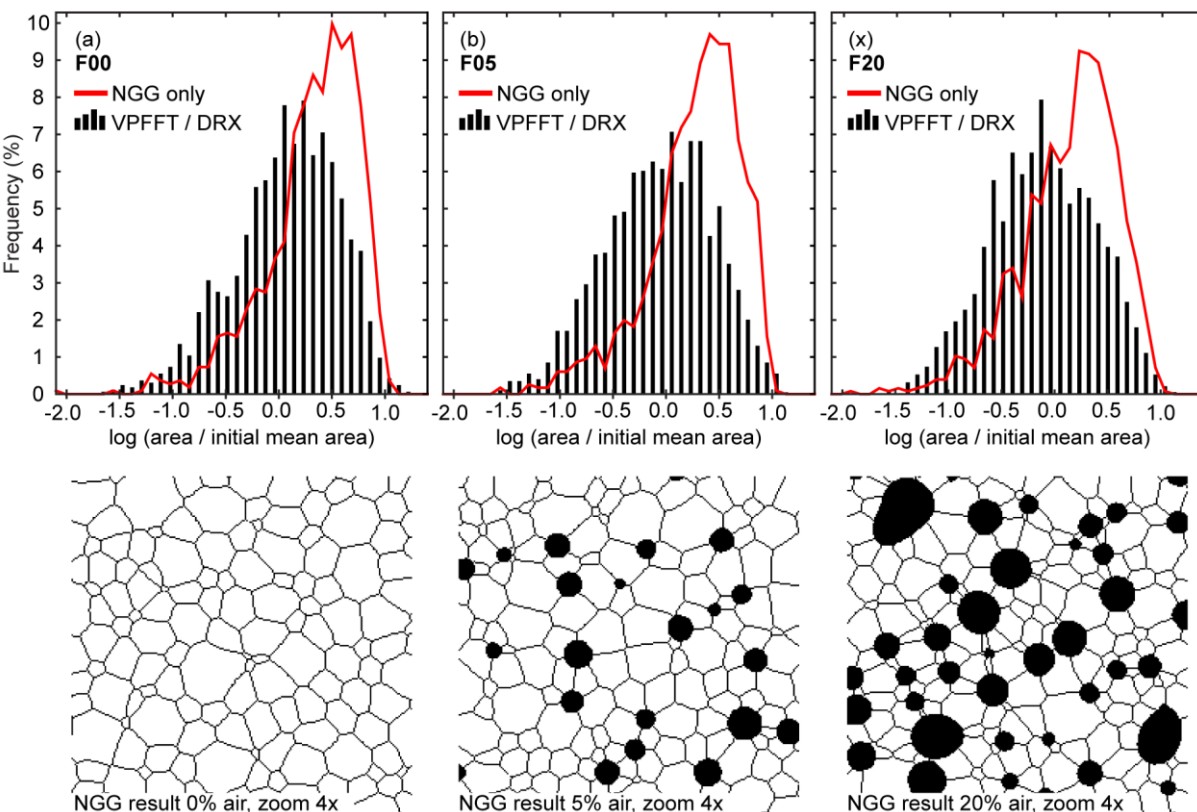

**Figure 6: Comparison of ice grain area histograms of the last simulation step of simulations (a) F00, (b) F05 and (c) F20. Normal grain growth (NGG) simulations using F00, F05 and F20 input models are displayed with the respective deformation and dynamic recrystallisation (VPFFT / DRX) simulation results. Areas were normalized to initial mean values that plot at a value of 0.0 on x-axis. For reference, a fourfold zoom in the resulting microstructures from NGG simulations is displayed below the histograms.**

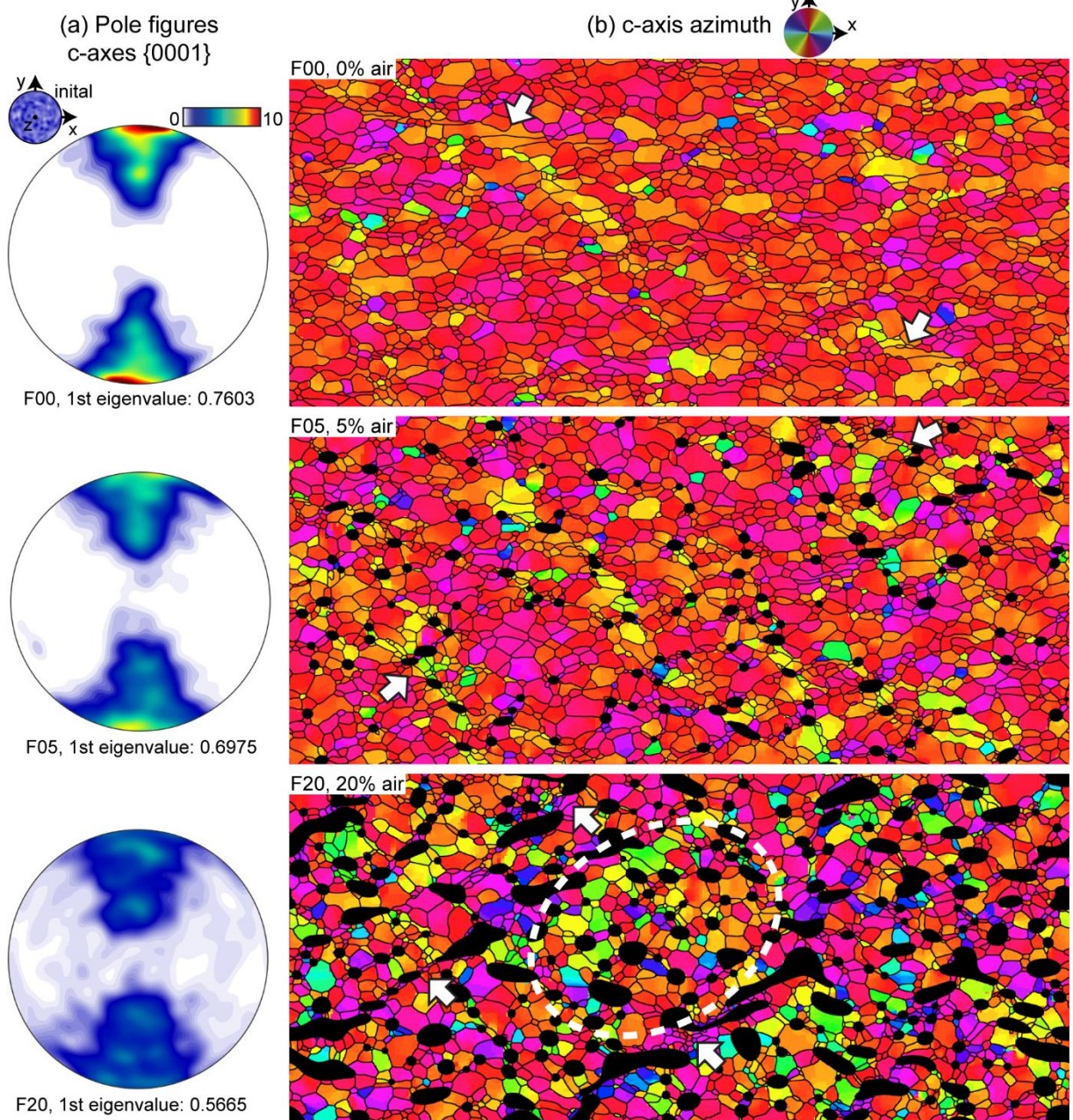

**Figure 7: Overview of c-axis orientations at 53 % vertical shortening for all simulations. (a) Pole figures with the projection plane parallel to the *x-y* plane of the 2D model. (b) Maps of c-axis azimuth distributions. Air inclusions are shown in black. White arrows in both images indicate zones of marked strain localisation. Dotted white line indicates zone of low strain where more random crystallographic orientations are preserved.**

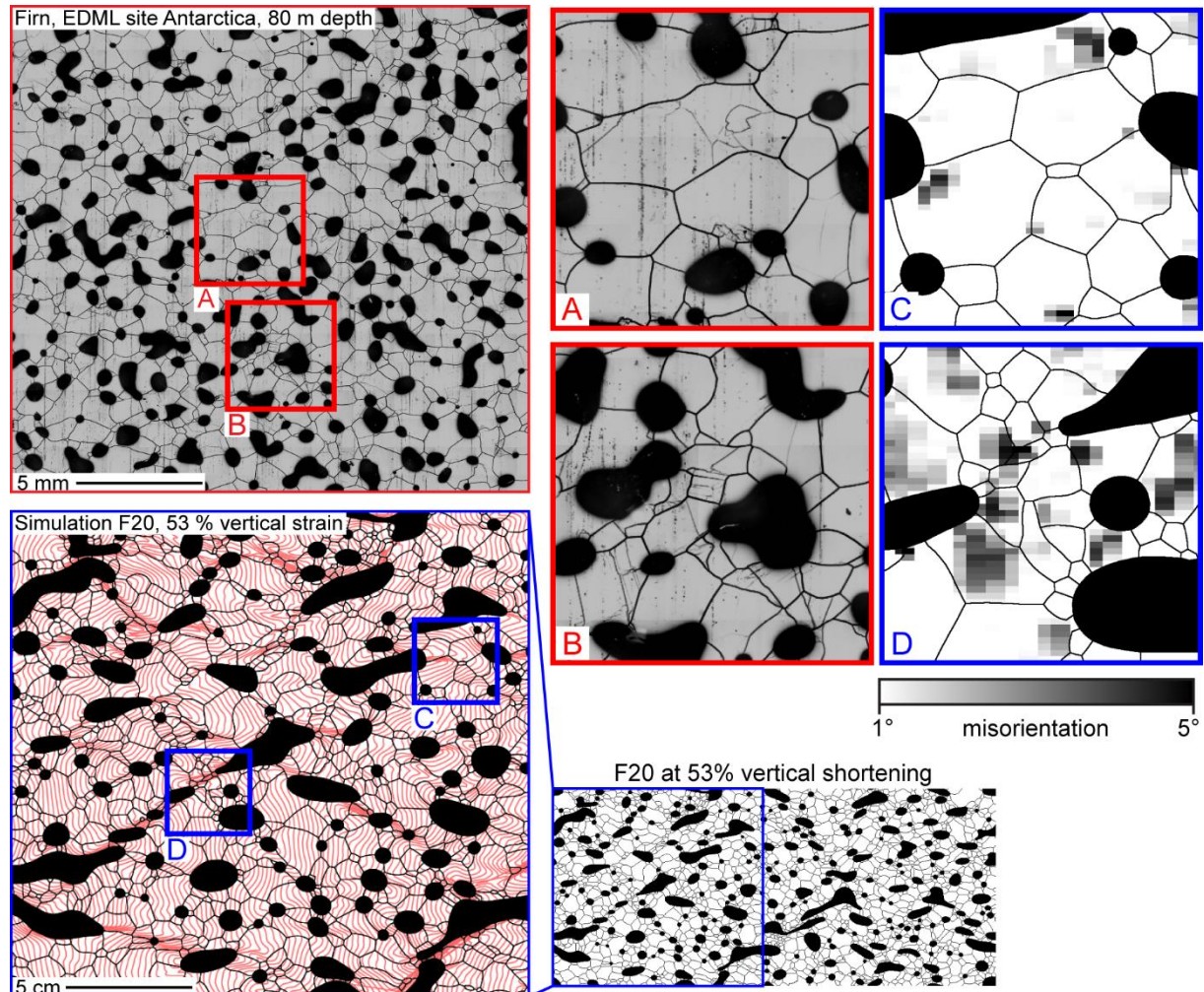

**Figure 8: Comparison of simulation results of F20 at 53% vertical shortening with firn microstructure mapping images from the EDML ice core site at 80 m depth (courtesy of Sepp Kipfstuhl). The detail areas A-D illustrate different microstructures occurring in relation to strain localisation. Grain boundaries stand out as black lines and subgrain boundaries are visible as fainter grey lines. Vertical stripes appearing in the overview image are related to the sample polishing technique and not reflecting any microstructural property. Greyscales C and D indicate local average misorientations as stored in *unodes* and therefore appear blurred due to the strong magnification. Note however, that strain paths of the natural microstructure (compaction) and the simulation (pure shear) are not identical and the comparison is restricted to the microstructural similarities and inferred localisation and recrystallisation processes.**

