# Peer review of "Strain localisation and dynamic recrystallisation in the ice-air aggregate: A numerical study"

_The Cryosphere, 2016_

## Referee Comment (RC1) · Anonymous Referee #1 · 25 Sep 2016

General Comments:

This paper presents micro-dynamical simulations of polycrystalline hcp ice containing air bubbles using a numerical approach based on the coupling of the numerical platform Elle, a front-tracking formulation that accounts for microstructure evolution due to different dynamic recrystallization processes (normal grain growth, strain-induced grain boundary migration, recovery, polygonization) and a viscoplastic model based on Fast Fourier Transform (VPFFT) to calculate the micromechanical fields (stress, strain-rate, velocity, etc.) due to deformation of the constituent ice crystals by dislocation creep. In particular, the stored energy field calculated with VPFFT provides the driving force for the aforementioned recrystallization processes. Details of the integration between Elle and VPFFT and applications to different geomaterial systems have been reported in previous papers by the same team, including studies of the micro-dynamics

of fully-dense ice polycrystals. This paper presents a new application of Elle/VPFFT, to study the ice-air system, aiming at a better understanding of the onset of dynamic recrystallization in, e.g. firn. One of the main conclusions with glaciological relevance of this study is that the presence of air bubbles induces a "composite material" behavior, which contributes to: a) higher strain localization in the ice crystals, and therefore faster onset of dynamic recrystallization compared with fully-dense ice, and b) weaker CPO–not caused by grain-boundary sliding–, as the volume fraction of air increases.

Specific Comments:

While the proposed approach is sound and the main conclusions are reasonable, the specific treatment of the air inclusion phase, as far as its constitutive behavior and the treatment of the ice-air interfaces are concerned, needs to be better explained, including a better disclosure of the approximations involved. It is reported (section 2.6) that air bubbles are represented by an incompressible crystalline material with the same crystallography and slip systems as for ice, and that tau_s-air is set 5000 times smaller than tau_basal of ice. This seems to imply (more clarity is needed here) that the air is represented by hcp crystals deforming by basal, prismatic and pyramidal slip with equal critical resolved stresses. In turn, this implies the somehow unrealistic assumptions of: a) "anisotropic" bubbles (although the anisotropy remains small) and b) a unit cell unable to accommodate any volume change. The first approximation could have been avoided by adopting an isotropic viscoplastic behavior, or, even better, imposing zero stiffness (i.e. vanishing stress) to the unodes belonging to the air phase. With this said, I suspect (but I'd like to hear this from the authors!) that the predictions would not be dramatically affected.

The second approximation is more delicate. The air phase/unit cell incompressibility implies the inability of the present approach to consider volume changes that are inherent to ice flowing under its own weight. Moreover, the incorporation of a constitutive description admitting compressibility would have also allowed improving the convoluted treatment of the behavior of the ice-air interfaces described in section 2.4.1, accounting explicitly for the effect of the bubbles' internal pressure, both in terms of mechanical behavior and as a controlling factor of the recrystallization process. Furthermore, the shortcomings associated with the simplified treatment of the air bubbles as an incompressible phase may be responsible for the somehow puzzling results, in the cases of the F05 and F20 microstructures, showing the overall porosity almost unaltered after ~50% vertical shortening. This makes the comparison with the EDML ice core at 80m depth presented in Fig. 8 questionable. This needs to be acknowledged and further model improvements to mitigate these limitations be discussed, before this paper is accepted for publication in The Cryosphere.

---

## Referee Comment (RC2) · Anonymous Referee #2 · 14 Oct 2016

Full-field numerical analysis of high density cold firn.

This paper aims to study the competing processes of normal grain growth, polygonization, and migration recrystallization in cold firn (very close to the density of ice). Their 2D model shows the importance of strain localization triggered by air inclusions that gives rise to locally high strain energies that drive grain boundary migration.

Overall, this paper is well written and I think this is a valuable analysis and the expressions of strain localization apparent in the model show that there is much more heterogeneity in recrystallization process in ice than we usually assume.

My primary concern with this paper is that some of the assumptions make it difficult, at best, to compare to the EDML firn thin sections. That does not say that the simulations aren't valuable, I just think the authors need to be upfront and clear from the beginning

what the simplifying assumptions are and how that affects their comparison. They state in the last sentence of the discussion that the comparison should be taken only approximately, but up until then they give the impression that they believe it is a very valid comparison.

In particular:

1. 2D versus 3D

2. The size and distribution of air pockets is set for each run but no information is given as to why those values were chosen. I would expect more smaller air pocket to behave differently than fewer larger air pockets, even with the same percentage of air. And visually, it looked like too few air pockets (although perhaps they assume the ice has undergone significant grain growth before reaching their initial microstructure).

3. The model air in the mixture is incompressible (yet, in real firn, it is highly compressible).

4. The stress regime is designed to generate 50% compressive strain along the vertical axis, this is accommodated by extension laterally (i.e. pure shear). The firn at EDML may have undergone 50% compressive strain along the vertical axis, but this was accommodated by densification including compression of air pockets (i.e. uniaxial compression). These are two very different stress regimes.

5. The grain size is HUGE compared to real polar plateau firn. This is not really discussed until the very end of the discussion. I had a difficult time figuring out why they started with such large crystals. Similarly the increments of 1% strain are high, how does this large crystals with fast strain rates affect the solution - does this prevent recovery from acting?

The overall conclusions of the model as it is presented are still valid. I would prefer to see the following changes:

1. Better explaining the effects of assumptions 1-5 above on the comparison between

model and real firn.

2. Either provide additional models that show better how the size of air pockets affects the results or at the very least, discuss the effect of this assumption on the results.

3. Either provide additional models that allow for some compression of the air, which might minimize some of the strain localization if deformation of the air can accommodate the changes needed in the ice crystals. Or, minimally, discuss the effects of this assumption on the solution.

Specific Comments:

Abstract: P1 Line 9/10: The first two sentences seem redundant.

Introduction P2 Line 1: "very" doesn't add anything

Line 13 and 16 - I am puzzled by some citations, Treverrow did not discover that CPO causes anisotropy, Montagnat was not the first to describe the effect on flow. While these are good papers, please cite papers that added to the discussion (and tell me why they added). These papers probably should be cited, but there are many more that have also contributed specific new ideas to the discussion, so please be specific as to why you have chosen those papers. There is a similar issue on page 3, Lines 29 and 30 - These two papers were not the first to describe folding in ice sheets due to anisotropy.

P3 Line 9: operate - present tense (please check all tenses).

P4 Line 28: accommodated

P6 Line 1 - redundant

Line 8 - How did you determine c and Mo and several of the parameters? I'm not sure I saw much in the way of a sensitivity study on the effect of variations in the parameters.

Line 15 - I think you can explain this a little more. Provide a little information as to why

you selected those values - more than just the citation, so we don't have to go read the other papers.

Lind 30 - "Recover be" ?? perhaps "by"

P7 Line 30 - "on" the results

P8 Line 2 - why was 0,5, and 20 chosen, especially considering that 10% is a commonly assumed volume of air for the bubble close-off depth?

Line 8 - repeated value for c, but still no explanation for that value.

P10 Line 13 - I realize that I haven't look at firn microstructure as much as Sep Kipfstuhl, but I was not under the impression that air pockets coalescing was a commonly occurring process in polar firn. I typically think of the pockets compressing and getting pushed to trip junctions, but not coalescing.

P11 Line 30 - "Apart from the scale difference... " This is where I have issue with the comparison. This very qualitative comparison for two very different systems seems strange (yes apples and oranges are both round and about the same size, so do we assume they are the same?). I don't argue that there are likely some of the same processes going on, I just don't think the comparison is done in a rigorous enough way. If the authors want to maintain this subjective comparison, it might be best shifted to the discussion section, than the results section, even better in a special part of the discussion section, so that it is clear that a direct rigorous comparison is not possible because of the assumptions, but it is still valuable to visually look. That kind of comparison does NOT belong in the results section.

P13 Line 15-20 - I had always understood that dynamic recrystallization was possible everywhere given strain energies, but is a much more dominant process above -10 (an activation energy transition point). Line 30 - "the initiation of this process is not only temperature dependent" - I'm not sure that anyone ever said that it's "initiation" was "only" temp dependent? In larger scale modeling is it much easier to parameterize the

migration recruits as being temp dependent, but this is a parametrization commonly used. Because the authors don't provide any comparison models, it is hard to tell how "dominant" the process is in -30 firn versus -10 firn. My main concern here is that they state that strain rate controls dynamic recrystallization as if that were a new idea.

This last statement would be more compelling if they presented a sweet of models at different temperature and different stress regimes both with and without the dynamic recrystallization process - to be able to show the effects of this process being active or not. Without any comparison simulations, it is hard to show what the effects are.

P14 Line 5-8 - this discussion about experimental strain rate and grain size selection should be up in methods (or maybe results), not in the discussion.

Line 10 - this should be in the methods section

Line 19 - specify what "it" is to be clear here

Line 25 - awkward sentence structure, please rewrite.

Line 33 - less, not lower

P 15 Line 14-15 - this statement should be early on in manuscript, or at least at the beginning of a discussion section about the comparision, not as an afterthought.

Line 17-25 - A conclusion should be used to talk about the these results in the context of larger questions. This is rather short conclusion that just repeats what has already been said. Please add some kind of bigger picture context. Why is it important to recognize that migration recrystallization happens (although slowly) in the firn? what can we do with this information in the future?

Table 1 - There is no discussion of the sensitivity of the model results to the selected parameters. Please provide some information.

Table 2 - just to reiterate when I saw this table, I was shocked at how large the grains were, the discussion of grain size is buried deeply in the discussion, please bring it up

front.

Figure 1 - I like this figure!

Figure 2 - I also like this figure, nice job explaining the components of the model.

—————————————————

---

## Author Comment (AC1) · 11 Nov 2016

Dear Dr. Ritz,

thank you very much for considering our manuscript for publication in The Cryosphere (IPICS special issue). We considered and addressed the referees' comments, replied to all of them and indicated specific changes that are made in the revised manuscript. The major changes in the revised manuscript are:

1) New discussion sup-chapter "4.5 Limitations of the modelling approach". In particular, the assumption of an incompressible air phase concerned the referees. We regard this an important issue and created a detailed reply that is found at the end of both reply letters.

[Figure]

2) New discussion sub-chapter "4.2 Natural firn microstructures and numerical simulations", the comparison has moved from section 3. To avoid redundancy, section 2.7.3 has been removed, and is now part of 4.2.

3) Supplementary figures S1 and S2 provide more simulations. The figures are also found with the specific replies.

Two additional changes were (a) the correction of a small error affecting the eigenvalues in Table 2 and Fig. 7, that does not affect the observations and interpretations. (b) After submission, Cyprych et al. published a study that supports our interpretation and adds to the discussion. We added Cyprych et al. (2016) as a reference and refer to the study in the revised manuscript.

The reply letters and a revised manuscript indicating the specific changes were uploaded to the discussion forum as the following files:

- AuthorReply_Referee1.pdf

- AuthorReply_Referee2.pdf

- RevisedManuscript_ChangesHighlighted.pdf

Thank you very much for your efforts.

With best regards,

Florian Steinbach and co-authors.

---

## Author Comment (AC2) · 11 Nov 2016

**Reply to referee comments by Anonymous Referee #1 on TC-2016-167**

**(Strain localisation and dynamic recrystallisation in the ice-air aggregate: A numerical study. F. Steinbach et al.)**

We thank Anonymous Referee #1 for a constructive review and regarding the proposed modelling approach and our conclusions as "sound and (…) reasonable". In the following, we reply to the specific concerns addressed by the referee and state the corresponding changes in the revised manuscript. The referee comments are cited in italics and our reply is in blue font. If not indicated differently, any reference to page or line numbers are with respect to the discussion paper, not the revised version.

*General Comments:*

*This paper presents micro-dynamical simulations of polycrystalline hcp ice containing air bubbles using a numerical approach based on the coupling of the numerical platform Elle, a front-tracking formulation that accounts for microstructure evolution due to different dynamic recrystallization processes (normal grain growth, strain-induced grain boundary migration, recovery, polygonization) and a viscoplastic model based on Fast Fourier Transform (VPFFT) to calculate the micromechanical fields (stress, strainrate, velocity, etc.) due to deformation of the constituent ice crystals by dislocation creep. In particular, the stored energy field calculated with VPFFT provides the driving force for the aforementioned recrystallization processes. Details of the integration between Elle and VPFFT and applications to different geomaterial systems have been reported in previous papers by the same team, including studies of the micro-dynamics of fully-dense ice polycrystals. This paper presents a new application of Elle/VPFFT, to study the ice-air system, aiming at a better understanding of the onset of dynamic recrystallization in, e.g. firn. One of the main conclusions with glaciological relevance of this study is that the presence of air bubbles induces a "composite material" behavior, which contributes to: a) higher strain localization in the ice crystals, and therefore faster onset of dynamic recrystallization compared with fully-dense ice, and b) weaker CPO–not caused by grain-boundary sliding–, as the volume fraction of air increases.*

We thank the referee for this concise general comment and summary of the main topics and conclusions of our paper. We are pleased to read that our objectives and conclusions are apparently clearly explained in the manuscript.

*Specific Comments:*

*While the proposed approach is sound and the main conclusions are reasonable, the specific treatment of the air inclusion phase, as far as its constitutive behavior and the treatment of the ice-air interfaces are concerned, needs to be better explained, including a better disclosure of the approximations involved. It is reported (section 2.6) that air bubbles are represented by an incompressible crystalline material with the same crystallography and slip systems as for ice, and that tau_s-air is set 5000 times smaller than tau_basal of ice. This seems to imply (more clarity is needed here) that the air is represented by hcp crystals deforming by basal, prismatic and pyramidal slip with equal critical resolved stresses.*

The implications the referee is deducing from our description in section 2.6 are correct. For numerical reasons, air was described as crystalline material (section 2.6, page 8, lines 13-14) with slip systems similar than in ice Ih (basal, prismatic and pyramidal slip). For ice Ih, the pyramidal and prismatic critical resolved shear stresses were 20 times higher than for slip on

the basal plane. The referee correctly assumes that the critical resolved shear stresses for all air slip systems were constant to approximate mechanical isotropy. As outlined in the manuscript, critical resolved shear stresses for the air phase were set 5000 times lower than for ice Ih basal slip. We agree that there is a need to better explain this approximation. Hence, we added a sentence on page 8, line 14. See our following reply for further explanations.

*In turn, this implies the somehow unrealistic assumptions of: a) "anisotropic" bubbles (although the anisotropy remains small) and b) a unit cell unable to accommodate any volume change. The first approximation could have been avoided by adopting an isotropic viscoplastic behavior, or, even better, imposing zero stiffness (i.e. vanishing stress) to the unodes belonging to the air phase. With this said, I suspect (but I'd like to hear this from the authors!) that the predictions would not be dramatically affected.*

We agree with the referee: Admittedly, this approach does not allow for a fully isotropic air material as deformation is only possible on defined slip systems. Our simplified approximation of the description of air inclusions is only correct if the results would not be dramatically affected by the unrealistic assumptions the referee was mentioning. Therefore, we updated our VPFFT code to impose zero stiffness to unodes in air inclusions (as suggested by the referee) to assume an air phase independent of any crystallography. For this comparison, the initial setups of F20 were used and deformed in one increment of pure shear to 1% vertical shortening to obtain instantaneous strain rates and stresses. We compared the results of the updated approach ("zero stiffness") with the VPFFT version used for the simulations presented in the manuscript (see attached figure R1.1). The results show:

(1) No significant difference in normalised von Mises strain rates, i.e. the location and intensity of strain localisation (cf. localisation factors "F")

(2) No significant difference in von Mises stresses for each unode obtained from the VPFFT output

(3) When using the VPFFT approach used for the simulations in the manuscript, mean pressure of both ice and air phase are zero. The variation in pressure in the air phase is very small and between one and two orders of magnitude smaller than in the ice phase.

One of the main observations of our study is that air inclusions cause and intensify strain localisation which provides driving forces for dynamic recrystallisation. The results of our comparison presented in figure R1.1 therefore indicate, that our treatment of the air phase as a very soft quasi-isotropic ice Ih crystal does not affect the predictions of the simulations. Using 5000 times softer slip for air "slip systems" than for ice basal slip and a stress exponent $n = 3$, we assume an even higher effective viscosity difference between the materials, which underlines why the results presented in figure R1.1 are essentially the same. As the results are almost identical, we did not re-run the simulations presented in the manuscript, but future simulations should use the new and updated VPFFT approach. Figure R1.1 will be provided as supplementary material.

In order to better discuss and present approximations made for the simulations (also the ones mentioned by referee #2), we added a paragraph to the new sub-chapter discussing model simplifications (4.5 Limitations of the modelling approach), where the simplified treatment of the air phase is discussed. We mention that for reasons outlined above our treatment of the air phase does not significantly affect the results.

[Figure]

(a) Instantaneous normalised von Mises strain rates

(b) von Mises stresses (dimensionless in VPFFT)

Air modelled as in
F00, F05 and F20
F = 0.509925

Air modelled imposing zero
stiffness to air unodes
F = 0.509947

Air modelled as in
F00, F05 and F20

Air modelled imposing zero
stiffness to air unodes

Figure R1.1: Comparison of the VPFFT model used for the simulations presented in the discussion paper with an modified algorithm that imposes zero stiffness to air unodes. These simulations comprised one step of VPFFT with an increment of 1% vertical shortening using the same initial setup as for simulation F20. We provide this figure as supplementary material and refer to it in the revised manuscript.

*The second approximation is more delicate. The air phase/unit cell incompressibility implies the inability of the present approach to consider volume changes that are inherent to ice flowing under its own weight. Moreover, the incorporation of a constitutive description admitting compressibility would have also allowed improving the convoluted treatment of the behavior of the ice-air interfaces described in section 2.4.1, accounting explicitly for the effect of the bubbles' internal pressure, both in terms of mechanical behavior and as a controlling factor of the recrystallization process. Furthermore, the shortcomings associated with the simplified treatment of the air bubbles as an incompressible phase may be responsible for the somehow puzzling results, in the cases of the F05 and F20 microstructures, showing the overall porosity almost unaltered after ~50% vertical shortening. This makes the comparison with the EDML ice core at 80m depth presented in Fig. 8 questionable. This needs to be acknowledged and further model improvements to mitigate these limitations be discussed, before this paper is accepted for publication in The Cryosphere.*

We thank the referee for pointing out an important assumption made in our simulation approach. Also referee #2 commented on the assumption that air is modelled as an incompressible material. In consequence, no porosity changes are possible during our simulations. The referees' concerns are clearly justified and correct. In the following we aim to better explain why we chose to use this assumption and discuss possibilities to mitigate this limitation. This reply can also be found at end of the reply to referee #2.

By imposing pure shear, we assume a deformation mode that conserves the total area of the simulation box, which does theoretically not allow for any volume change and implies conservation of mass for both phases. However, firn is characterized by most vertical shortening achieved by compaction of the pore space causing a significant air volume loss. In general, we would like to remark, that the evolution of our numerical microstructures cannot be regarded as an evolution with depth (as would be the case in natural firn and ice). In fact, the microstructure in each simulation step can be regarded as the microstructure that results from the deformation of a material with an unknown previous porosity to the actual situation. We refrain from any study of depth evolution of porosity, inclusion shape or distribution and

remark, that the scope of the manuscript is a study of deformation and recrystallisation processes within the ice at the presence of a very weak phase.

Theoretically, the compaction of a pore is a function of the surface energy driving inward bubble surface movement and the inner bubble pressure counter-acting this movement. The latter depends on parameters such as the overburden pressure, bubble shape and connectivity. In a state of equilibrium, a bubble's size is does not change implying static conditions. Since the simulations do not incorporate gravitational forces, overburden pressure is unknown and the theoretical "area energy" is used to counter act surface energy (cf. section 2.4.1, equations (3) and (4) and Roessiger et al., 2014). The pre-factor $c$ can be regarded as an approximation of a compressibility factor that controls how quickly this equilibrium is reached (Roessiger et al., 2014). The lower the factor $c$, the less "area energy" is counter acting the surface energy that tends to decrease the overall cross sectional area of the bubbles. In turn, this means more cross sectional area change is allowed causing a stronger violation of the conservation of mass requirement.

To fulfil the conservation of mass requirement in our simulations, any movement of the ice-air interface that is not mass conserving should actually be inhibited. This would however lead to complete "freezing" of the interfaces, an even more unrealistic assumption. Therefore, we allow movements of the ice-air interfaces that preserve the overall porosity, but still allow for sufficient shape changes of the bubbles. Preparatory tests yielded $c = 0.1$ as a compromise to achieve this. With this, we use a 10 times higher factor $c$ than Roessiger et al. (2014), who modelled static conditions without deformation.

The current VPFFT code does, unfortunately, not include a compressible phase or voids. This is not an intrinsic limitation of the model, and a version without this limitation is under development. The current model is, therefore, not capable of simulating compaction, and we limited ourselves to area-conservative pure shear. Admittedly, this raises questions on the comparison of the simulations with the EDML firn image (Fig. 8). In the revised manuscript, we discuss the limitations associated with assuming incompressibility and explicitly highlight, that the comparison with the firn image has to be taken qualitatively and as a comparison of inferred processes and their expression in the microstructure.

Specific actions taken in the revised manuscript:

1. As a reaction to both referees' concerns, we created the new sub-chapter "4.5 Limitations of the modelling approach" to discuss approximations made in our simulations. A condensed version of the explanations above is part of this chapter.

2. The role of the pre-factor $c$ and our choice of $c = 0.1$ is now better explained in section 2.4.1 and in the new section 4.5.

3. The comparison with the EDML firn image (Fig. 8) has moved to another new sub-chapter in discussion (section 4.2). We present the natural firn image as a first qualitative comparison with an Elle/VPFFT simulation on ice microdynamics. The intention of this comparison is trying to identify processes observed in the simulations also in natural firn (i.e. strain localisation in the vicinity of bubbles associated with enhanced dynamic recrystallisation). The limitations caused by the modelling approach are discussed and we state that we refrain from any quantitative comparison.

---

## Author Comment (AC3) · 11 Nov 2016

**Reply to referee comments by Anonymous Referee #2 on TC-2016-167**

**(Strain localisation and dynamic recrystallisation in the ice-air aggregate: A numerical study. F. Steinbach et al.)**

We are thankful for the useful und constructive review of referee #2 in which our paper is regarded as "well written" and the analysis as "valuable". In the following, we reply to the specific concerns addressed by the referee and state the corresponding changes in the revised manuscript. The referee comments are cited in italics and our reply is in blue font. If not indicated differently, any reference to page or line numbers are with respect to the discussion paper, not the revised version.

*Full-field numerical analysis of high density cold firn.*

*This paper aims to study the competing processes of normal grain growth, polygonization, and migration recrystallization in cold firn (very close to the density of ice). Their 2D model shows the importance of strain localization triggered by air inclusions that gives rise to locally high strain energies that drive grain boundary migration.*

*Overall, this paper is well written and I think this is a valuable analysis and the expressions of strain localization apparent in the model show that there is much more heterogeneity in recrystallization process in ice than we usually assume.*

*My primary concern with this paper is that some of the assumptions make it difficult, at best, to compare to the EDML firn thin sections. That does not say that the simulations aren't valuable, I just think the authors need to be upfront and clear from the beginning what the simplifying assumptions are and how that affects their comparison. They state in the last sentence of the discussion that the comparison should be taken only approximately, but up until then they give the impression that they believe it is a very valid comparison.*

We are thankful for that helpful advice. Referee #1 raised similar concerns and we therefore decided for some general changes in the manuscript (details are outlined within the replies). In the following we summarize these changes and give some general comments:

(1) The comparison between an EDML firn core image and simulation results has moved in a new sub-section in discussion (new section "4.2 Natural firn microstructures and numerical simulations" in revised manuscript). To avoid redundancy, section 2.7.3 was removed, and the condensed content is found in the new section 4.2. The comparison intends to provide a first comparison of an Elle/VPFFT model of ice micro-dynamics to natural samples. The comparison is designed as a purely visual and qualitative comparison that compares processes observed in the simulation and suspected in natural firn (namely the process of strain localisation in the vicinity of bubbles and the related increased strain energies driving dynamic recrystallisation).

(2) We created another new sub-section in discussion in the revised manuscript ("4.5 Limitations of the modelling approach"). Here, we discuss in more detail the approximations and limitations of the approach and discuss possible solutions for future studies.

*In particular:*

*1. 2D versus 3D*

Correct, the stereological effect when comparing 2D models with 2D slices from a 3D sample is important. A theoretical circular grain (or bubble) in our simulations can be imagined to continue as an infinite cylinder in the 3rd dimension. Hence, any cross sectional area in a 2D plane cut parallel to the modelling plane will be the maximum cross sectional area possible. This is the major difference between our 2D simulations and a 2D cut through a 3D sample of natural ice. In a natural 2D sample, the cross sectional area of a grain is to a certain probability not the maximum cross sectional area of this grain (e.g. see the work of Anderson et al. (1989) in Philosophical Magazine B). Essentially this means that our simulations exhibit the maximum grain areas possible, however, a 2D cut through a 3D natural sample tends to underestimate the real grain sizes. In order to do a proper grain size comparison between simulation and natural sample (but we refrain from doing this), corrections on the natural 2D grain size would be necessary.

The abovementioned is summarized in the new sub-section in discussion (4.2 in revised manuscript) that compares the EDML firn image with modelling results.

*2. The size and distribution of air pockets is set for each run but no information is given as to why those values were chosen. I would expect more smaller air pocket to behave differently than fewer larger air pockets, even with the same percentage of air. And visually, it looked like too few air pockets (although perhaps they assume the ice has undergone significant grain growth before reaching their initial microstructure).*

The distribution of air inclusions was designed by using an initial setup of pure ice and setting air properties to random ice grains. We hoped this to cause the most realistic bubble distribution. The number of grains was chosen to later (after applying surface energy boundary migration) produce the desired fractions of air (0, 5, 20%). Afterwards, surface energy based grain boundary migration was applied causing the air bubbles to adopt more realistic (circular) shapes. It is correct, that there is a lower number of air inclusions in the simulation image than in the presented EDML firn image. We can expect that the size and distribution of air inclusions does not significantly change the processes interpreted from the results (localisation and locally high strain energies). Some more comments on this:

- The main effect of the absolute size of the bubbles is on their ability to maintain a circular shape. Small bubbles are more rounded than large ones in the model (see figure R2.1) and predicted by theory (discussed, for example, in Walte et al. 2011. Earth and Planetary Science Letters 305, 124-134). The shape of the bubbles is comparable to those observed in firn and bubbly ice.

- The size of bubbles relative to that of ice grains is possibly more important than the absolute size of bubbles. This determines the fraction of grain boundaries influenced by nearby bubbles and those further away (Roessiger et al. 2014. Journal of Structural geology 61, 123-132). A systematic study of this effect was not undertaken here, as the work focussed on the localisation behaviour induced by the presence of bubbles.

- The strain localising effect of air inclusions giving rise to locally higher strain energies driving dynamic recrystallisation is hardly affected by the size and distribution of bubbles: We did systematic tests on how the distribution of bubbles affects this general observations in the manuscript (see figure R2.1). The tests are comprised of two simulation setups, one with many small inclusions, another one with a low amount of large inclusions. The overall fraction of air is constant in both setups and the same as in setup F20. The results show that strain localisation in the sense as described in the manuscript is still occurring with many small and a few large air inclusions.

We added figure R2.1 as a supplementary figure (S1) showing the results of the mentioned trial simulations. Furthermore, we acknowledge that comparing size, shape and distribution of bubbles with natural samples is not possible and not our aim.

[Figure]

Figure R2.1: Comparison of simulations varying the distribution of air inclusions. The simulations were performed under the same conditions as for simulation F20, which is presented in the manuscript. The results show that strain localisation and associated locally higher strain energies are occurring in both setups. A notable difference is that large bubbles cannot maintain their approximately circular shape. We provide this figure as a supplementary figure and refer to it in the revised manuscript.

*3. The model air in the mixture is incompressible (yet, in real firn, it is highly compressible).*

We agree with the referee; this is an important point that needs more clarification as it has also been mentioned by referee #1. Since we regard this clarification as important, we created a detailed reply that is found at the end of this document. Please find more explanations in this section.

*4. The stress regime is designed to generate 50% compressive strain along the vertical axis, this is accommodated by extension laterally (i.e. pure shear). The firn at EDML may have undergone 50% compressive strain along the vertical axis, but this was accommodated by densification including compression of air pockets (i.e. uniaxial compression). These are two very different stress regimes.*

The stress regimes in simulation and firn are indeed very different. We are unfortunately currently not able to model a stress regime more similar to the one observed in firn. Our assumption of assuming an incompressible air phase relates to this deformation mode that is not allowing for area changes (see specific reply at the end of this document). In the revised manuscript, we discuss this point when comparing to EDML firn (section 4.2).

*5. The grain size is HUGE compared to real polar plateau firn. This is not really discussed until the very end of the discussion. I had a difficult time figuring out why they started with such large crystals. Similarly the increments of 1% strain are high, how does this large crystals with fast strain rates affect the solution - does this prevent recovery from acting?*

Admittedly, the chosen initial and resulting final grain size is large in comparison to natural firn grain sizes. Visually, the natural firn image and simulation result are comparable, however, the actual grain size of the model is about an order of magnitude higher and the strain rate is faster (cf. 2D/3D effects outlined before).

We decided for this large initial grain sizes mainly to make our simulations comparable to the ice simulations without air bubbles by Llorens et al. (2016a,b). Preparatory sensitivity tests showed that final grain sizes strongly vary with parameters such as mainly mobilities, critical high angle grain boundary angle for polygonisation or energy per dislocation line. We chose to use literature values for these parameters to allow comparability with previous modelling instead of introducing new values for these parameters. Unfortunately, with these given input parameters, the simulations predict larger grain sizes than expected from firn and the fast input strain rate. We would like to remark, that although rigorous experimental studies have been performed to determine some of the material properties (e.g. by Nasello et al. (2005) on mobilities), we cannot be sure of how applicable those are for numerical models. Presumably, especially the numerical mobilities need proper adjustment to quantitatively compare future simulations to natural microstructures. Furthermore, the effect of micro- to nanoparticles is suspected to decrease boundary migration and hence boundary migration rates, but is currently not incorporated in the numerical approach. The discrepancy in grain size actually shows that the modelling can be used to gain better estimates of often poorly constrained values of a range of parameters, such as grain-boundary mobility, dislocation energy, etc. This was, however, not the aim of this study.

About the "high" increments of 1% strain together with large crystals and strain rate: This increment is actually the lowest one so far used in this kind of models on ice microdynamics. Simulations by Llorens et al. (2016a,b) and Jansen et al. (2016) use increments at least twice as high. Lower increments lead to error reduction, but increase computation time. Currently increments of 1% provide the best compromise between both. Future code refinements aim to allow smaller increments. Recovery is not prevented from acting, but still occurring associated with polygonisation, which is visible in the simulation videos (supplementary material).

We remark that large grain sizes only affect the Elle based recrystallisation approach. The VPFFT approach is dimensionless and is only affected by a lower resolution limit, which is determined by the number of *unodes* in the simulation. As strain localisation is caused by the distribution of viscoplastic deformation provided by the VPFFT code, we cannot expect a strong effect of grain size controlling strain localisation and the balance of driving forces for static and dynamic recrystallisation. In fact, as VPFFT is dimensionless and still provides strain localisation bands in any kind of setup, future research should investigate if strain localisation in ice is a scale independent process.

In the new sub-chapter 4.2, we acknowledge that our grain sizes are only qualitatively comparable with natural firn. We also added a discussion of the numerically predicted grain sizes to the new sub-chapter "4.5 Limitations of the modelling approach".

*The overall conclusions of the model as it is presented are still valid. I would prefer to see the following changes:*

*1. Better explaining the effects of assumptions 1-5 above on the comparison between model and real firn.*

We created a new sub-chapter in discussion (4.2) to compare to EDML firn and include a discussion of these assumptions.

*2. Either provide additional models that show better how the size of air pockets affects the results or at the very least, discuss the effect of this assumption on the results.*

We performed trial simulations to test how the size and distribution affects the results and, more specifically, the conclusions drawn (see figure R2.1). The total fraction of air in these tests is constant and the same as in setup F20 (like the remaining input parameters). While the 1$^{st}$ setup has a high number of small bubbles, the 2$^{nd}$ setup has a low number of large bubbles. Strain localisation between the air inclusion that produces locally high strain energies driving dynamic recrystallisation is observed in all these simulations.

We now provide figure R2.1 as a supplementary figure (S1) and mention it in the revised manuscript.

*3. Either provide additional models that allow for some compression of the air, which might minimize some of the strain localization if deformation of the air can accommodate the changes needed in the ice crystals. Or, minimally, discuss the effects of this assumption on the solution.*

Unfortunately, the current numerical setup is not able to correctly include the effect of compaction, please find more explanations on this topic in a detailed reply at the end of this document. We agree that the effects of our assumption at least need to be discussed in more detail. This discussion and an explanation of ways to mitigate this limitation in future studies is part of the new sub-chapter "4.5 Limitations of the modelling approach".

*Specific Comments:*

*Abstract: P1 Line 9/10: The first two sentences seem redundant.*

We understand the redundancy the referee sees in this expression and are thankful for this hint. For clarification: The first sentence was intended to briefly introduce what we did. With the second sentence we wanted to highlight that we are the first to present and use such a modelling approach for dynamic recrystallisation and deformation in ANY polyphase crystalline aggregate (including rocks, metals, etc.). To avoid any redundancy and use a more precise expression, we changed the second sentence of the abstract.

*Introduction P2 Line 1: "very" doesn't add anything*

We agree. The word "very" has been removed.

*Line 13 and 16 - I am puzzled by some citations, Treverrow did not discover that CPO causes anisotropy, Montagnat was not the first to describe the effect on flow. While these are good papers, please cite papers that added to the discussion (and tell me why they added). These papers probably should be cited, but there are many more that have also contributed specific new ideas to the discussion, so please be specific as to why you have chosen those papers.*

We thank the referee for this hint.

We chose to use to cite this literature on CPO causing mechanical anisotropy since Budd and Jacka provided an overview in their review paper and Treverrow et al. is a more recent work supporting that CPO causes anisotropy. Montagnat et al. (2011) was used as a more recent study that (a) explicitly mentions that recrystallisation influences the flow of ice (b) using two both numerical modelling and experimentally deformed ice. However, we see that we should choose

more suitable literature here and be specific about what we want to express. Out specific changes are:

- We now also refer to the early experimental work by Steinemann (1954) and Gao and Jacka (1987) and explain that these studies contributed to our knowledge about CPO causing large-scale mechanical anisotropy.

- We omitted the reference to Montagnat et al. (2011). We want to highlight that recrystallisation affects fabric developments, which is in turn affecting the creep behaviour. We therefore now cite three papers that contributed to how rotation recrystallisation and (static and dynamic) grain boundary migration affect fabric development and hence the creep behaviour (Duval and Castelnau, 1995; Castelnau et al., 1996; Duval et al. 2000). We changed the sentence on page 2 line 15.

*There is a similar issue on page 3, Lines 29 and 30 - These two papers were not the first to describe folding in ice sheets due to anisotropy.*

The referee is correct. Those two papers are not the first to describe folding in relation to anisotropy. However, this is not what we intended to express in this sentence: In fact, the whole paragraph is intended to give the reader a brief overview on the Elle platform and how it has been applied in similar studies (cf. beginning of the sentence: "Recent applications of methods (…) are (…)."). We chose to cite these two papers at this point because they represent the first application of Elle / VPFFT on the topic of folding in ice sheets. With respect to this intention of the paragraph, the citations should be suitable and valid.

We are still thankful to the referee for pointing out this issue, apparently we need to make more clear why we chose to use these citations. As a reaction, we changed the beginning of the sentence (page 3 line 27) to be more clear about the intention of this sentence.

*P3 Line 9: operate - present tense (please check all tenses).*

Thank you, we changed this. All tenses have been double checked.

*P4 Line 28: accommodated*

Thank you for the hint. We changed this.

*P6 Line 1 – redundant*

We agree with the referee. The sentence on page 5 line 26-27 already mentioned that the stored strain energy was not taken into account for ice-air boundaries. We changed the sentence on page 6 line 1 accordingly.

*Line 8 - How did you determine c and Mo and several of the parameters? I'm not sure I saw much in the way of a sensitivity study on the effect of variations in the parameters.*

Most parameters were chosen according to published literature (wherever a reference is given). Numerous sensitivity tests took place during the development of the model. In summary, they show how sensitive the model is to changes in mainly the grain boundary mobility or other parameters such as the assumed high angle grain boundary angle. Hence, most of the parameters were set to the values of either previous numerical modelling (to allow comparability), experimental work or other literature values. We did not want to introduce new values differing from previous studies as this would require more detailed sensitivity tests, which is beyond the scope of the paper. In addition, we chose literature values as they mostly resulted in low numerical errors and hence a stable simulation.

An example for a parameter that is not given by the literature is the factor *c* (section 2.4.1). We chose to assume equilibrium between surface energies shrinking the bubbles and inner bubble pressures counter-acting the shrinkage. In turn, this causes our simulations to assume

incompressibility for the modelled box (please find more information in our detailed replyon the incompressibility assumption and in new sub-chapter 4.5). To achieve this equilibrium during the simulations, preparatory tests yield that $c=0.1$ is a good compromise that still allows the bubbles to maintain a realistic (circular) shape, without causing unacceptable changes in air fraction.

In the revised manuscript, we now better explain why we use $c = 0.1$ after the paragraph ending on page 6 line 14 (in discussion paper). On page 7 line 18 of the discussion paper ($\alpha_{hagb} = 5$ °), we added a half sentence to express that we used this critical angle to be conservative. A smaller angle would lead to lower grain sizes as grain splitting by polygonisation is the most effective grain size reducing mechanism in our simulations. We expect that this would cause dynamic recrystallisation to become even more obvious as more small grains along high strain zones would develop.

*Line 15 - I think you can explain this a little more. Provide a little information as to why you selected those values - more than just the citation, so we don't have to go read the other papers.*

This relates to the previous comment. We changed this paragraph to provide more information about the mentioned parameters in this paragraph. For further explanation, see our reply to the previous comment.

*Lind 30 - "Recover be" ?? perhaps "by"*

Thanks, that has been changed

*P7 Line 30 - "on" the results*

Thanks again, we changed it.

*P8 Line 2 - why was 0,5, and 20 chosen, especially considering that 10% is a commonly assumed volume of air for the bubble close-off depth?*

The referee is correct: 10% porosity is usually assumed for the firn-ice transition. Our goal was not to investigate processes at the firn-ice transition only, but in a theoretical ice-air aggregate in general. For this reason, we chose three different porosities, 20% for the material in firn (above the firn-ice transition), 5% for bubbly ice well below the firn ice transition and 0% for comparison and reference. Additionally, our simplification of assuming an incompressible air phase does not allow us to relate our simulation to any specific depth (or porosity) in the ice or firn column, which is why we refrained from using "real firn" porosities.

We agree that this intention should become clearer in the manuscript and added three sentences on page 8 line 3.

*Line 8 - repeated value for c, but still no explanation for that value.*

There is obviously a need for us to be more specific here: This sentence explains how we introduced the air inclusions in the numerical microstructure. To create initial setups for F05 and F20, air conditions were set to a number of grains in setup F00 followed by applying surface energy based boundary migration, which allowed them to adopt a near 180° dihedral angle (i.e. a circular shape). For this purpose, we also needed to decide for a factor $c$ and for consistency used the same value as for the actual dynamic recrystallisation and deformation simulations. This is what we wished to express here. Pleased find a description on why we chose for $c = 0.1$ in the specific reply on incompressibility assumption that is provided at the end of this document.

To avoid that our expression can misleadingly be understood as a repetition, we changed the last sentences of this paragraph. An explanation why $c = 0.1$ was used is added (cf. our corresponding reply at the end of this document).

*P10 Line 13 - I realize that I haven't look at firn microstructure as much as Sep Kipfstuhl, but I was not under the impression that air pockets coalescing was a commonly occurring process in polar firn. I typically think of the pockets compressing and getting pushed to trip junctions, but not coalescing.*

On page 10 line 13, we did not want to express that bubbles coalesce as would be expected in firn, but instead we intended to state a basic observation on the simulations. Coalescence in simulations would be expected from simulations on static grain boundary migration in the ice-air aggregate by Roessiger et al. (2014). Evidently, we see that the large air inclusions in the simulations are created due to coalescence (see supplementary movies). Therefore, mentioning this basic observation in results section is correct.

Coalescence of bubbles is a function of deformation and surface-energy effects. It is observed in experiments on liquid (melt) pockets in deforming rocks and happens in the simulations. If the bubbles shrink faster than the convergence caused by deformation, coalescence would be suppressed. This may be the case in natural firn, but is difficult to ascertain. Due to the high surface energy, a new bubble that results from a merger very quickly regains a spherical shape. Bubble coalescence is therefore difficult to determine from bubble shapes only. Bubble size distributions may be a better (or only?) way to determine the importance of coalescence in natural samples (Roessiger et al. 2014).

We did *not* change the sentence on page 10 line 12, because the basic observation of coalescing "numerical bubbles" is correct (see supplementary video). Since this is the results section, any discussion on this topic would be inappropriate. But: We added a remark in the new sub-chapter 4.2 (comparison with EDML sample) that bubble coalescence may be less common in natural firn than in our simulations.

*P11 Line 30 - "Apart from the scale difference… " This is where I have issue with the comparison. This very qualitative comparison for two very different systems seems strange (yes apples and oranges are both round and about the same size, so do we assume they are the same?). I don't argue that there are likely some of the same processes going on, I just don't think the comparison is done in a rigorous enough way. If the authors want to maintain this subjective comparison, it might be best shifted to the discussion section, than the results section, even better in a special part of the discussion section, so that it is clear that a direct rigorous comparison is not possible because of the assumptions, but it is still valuable to visually look. That kind of comparison does NOT belong in the results section.*

We agree with the referee's concerns. As outlined by the referee, we wanted the comparison to be qualitative and purely visual. It intends to compare the processes going on and show, that both in simulation and (probably) in natural firn, strain localisation as a result of bubbles as a second phase occurs and leads to locally higher strain energies associated with dynamic recrystallisation.

As described in other parts of the results, we moved the comparison to a new sub-chapter in discussion (4.2 in revised manuscript). We state that the comparison is qualitative and discuss the assumptions made for the simulation. In the course of this, we moved and shortened the explanations of methods section 2.7.3 to the new sub-chapter. Furthermore, we avoid the expression "apart from the scale difference (…)".

*P13 Line 15-20 - I had always understood that dynamic recrystallization was possible everywhere given strain energies, but is a much more dominant process above -10 (an activation energy transition point). Line 30 - "the initiation of this process is not only temperature dependent" - I'm not sure that anyone ever said that it's "initiation" was "only" temp dependent? In larger scale modeling is it much easier to parameterize the migration recruits as being temp dependent, but this is a parametrization commonly used. Because the*

*authors don't provide any comparison models, it is hard to tell how "dominant" the process is in -30 firn versus -10 firn. My main concern here is that they state that strain rate controls dynamic recrystallization as if that were a new idea. This last statement would be more compelling if they presented a sweet of models at different temperature and different stress regimes both with and without the dynamic recrystallization process - to be able to show the effects of this process being active or not. Without any comparison simulations, it is hard to show what the effects are.*

The referee is correct that no one ever strictly said that the initiation of strain induced boundary migration is only temperature dependant. However, it has been proposed in earlier studies that "migration recrystallisation occurs for temperatures higher than -10°C" (Duval and Castelnau, 1995). This we interpret as that it is *most dominant* at temperatures of -10°C or higher. Our modelling, together with previous observations on firn by e.g. Kipfstuhl et al. (2009) or Faria et al. (2014b, Appendix B), shows that even at lower temperatures the strain energies can drive dynamic recrystallisation and that, therefore, strain induced boundary migration can be a significant or even dominant processes.

We thank the referee for pointing out this inaccurate expression. We changed the sentence at page 13 lines 15-17 and page 13 lines 29-31 to use similar expressions to those in our reply above and outline that Duval and Castelnau (1995) suggested that strain induced boundary migration is *most dominant* at temperatures of -10°C or higher.

*P14 Line 5-8 - this discussion about experimental strain rate and grain size selection should be up in methods (or maybe results), not in the discussion.*

Thank you for pointing this out. The statement on grain sizes has moved to section 2.6 (Methods: Simulation setup) mentioning that we used these grain sizes to be consistent with Llorens et al, (2016a,b). The statements on strain rates were actually redundant on page 14, lines 6-9, which is why we merged them and moved the statement to section 2.6. Effectively, this means we removed the contents on page 15 lines 3-11 (see reply to following comment).

*Line 10 - this should be in the methods section*

We agree. The paragraph has been condensed into three sentences and moved to the methods section 2.6.

*Line 19 - specify what "it" is to be clear here*

Thanks, this expression needs improvement. Llorens et al. (2016b) show that with recrystallisation active, it is difficult or impossible to see the actual strain distribution or strain localisation in the grain boundary network. Strong recrystallisation rapidly overprints any strained grain boundary, which makes the grain boundaries unsuitable as strain markers. The actual strain heterogeneity in ice is therefore higher than the boundary network may suggest.

What we mean here with "it" is strain localisation. We changed the sentence accordingly.

*Line 25 - awkward sentence structure, please rewrite.*

Thank you, we rewrote the sentence structure in the revised manuscript.

*Line 33 - less, not lower*

Thanks, we changed it.

*P 15 Line 14-15 - this statement should be early on in manuscript, or at least at the beginning of a discussion section about the comparision, not as an afterthought.*

We agree with the referee that this statement is important and underlines that the results have to be discussed with respect to the model approximations. We shifted this sentence in front of the discussion chapter on grain size analyses and accordingly changed the first paragraph. As

the comparison with the EDML firn image moved to the discussion, we also highlight the statement in the new sub-chapter 4.2. Furthermore, the new sub-chapter on limitations of the modelling approach (4.5) should clarify this even more.

*Line 17-25 - A conclusion should be used to talk about the these results in the context of larger questions. This is rather short conclusion that just repeats what has already been said. Please add some kind of bigger picture context. Why is it important to recognize that migration recrystallization happens (although slowly) in the firn? What can we do with this information in the future?*

We are thankful for this suggestion and took the opportunity to modify the conclusions by adding:

- Ice sheet deformation may be more heterogeneous than previously thought.

- Strain localisation is not the exception, but rather the rule in ice sheets and glaciers. Together with anisotropy, second phases (such as air bubbles) provide an effective mechanism for strain localisation.

- Localisation is a process that could be considered in future firn densification models

- Due to strain localisation, the rate of fabric change is locally higher. This may happen especially in firn, where bubbles are most abundant and can cause localisation.

- The used VPFFT code is dimensionless, future research could focus on the question whether strain localisation may generally occur on a range of scales in ice.

*Table 1 - There is no discussion of the sensitivity of the model results to the selected parameters. Please provide some information.*

The referee's comments on page 6 lines 8 and 15 expressed similar concerns, we kindly ask the referee and the reader to find our more detailed reply after these comments.

We agree with the referee, that this needs more clarification and we therefore changed the text in the corresponding parts in section 2 (see our specific reply to referee's comments on page 6 line 8 and 15) to contain more information on the selected parameters.

We now also refer the reader to section 2.4 and 2.6 in the table caption.

*Table 2 - just to reiterate when I saw this table, I was shocked at how large the grains were, the discussion of grain size is buried deeply in the discussion, please bring it up front.*

We agree with the referee, that the discussion of the large grain sizes needs more clarification, especially when it comes to a comparison with EDML firn images. As mentioned before, we moved this comparison to the discussion section (new sub-chapter 4.2) and discuss with respect to the numerical grain sizes. Please find more explanations in our reply earlier in this document on the general comments on assumptions made in the modelling approach (in particular assumption 5).

*Figure 1 - I like this figure!*

*Figure 2 - I also like this figure, nice job explaining the components of the model.*

We are glad about these kind of remarks.

Following page: See our reply to the referee's comment on our assumption of an incompressible air phase.

**Reply on referee's comments on assuming an incompressible air phase**

We thank the referee for pointing out an important assumption made in our simulation approach. Also referee #1 commented on the assumption that air is modelled as an incompressible material. In consequence, no porosity changes are possible during our simulations. The referees' concerns are clearly justified and correct. In the following we aim to better explain why we chose to use this assumption and discuss possibilities to mitigate this limitation. This reply can also be found at end of the reply to referee #1.

By imposing pure shear, we assume a deformation mode that conserves the total area of the simulation box, which does theoretically not allow for any volume change and implies conservation of mass for both phases. However, firn is characterized by most vertical shortening achieved by compaction of the pore space causing a significant air volume loss. In general, we would like to remark, that the evolution of our numerical microstructures cannot be regarded as an evolution with depth (as would be the case in natural firn and ice). In fact, the microstructure in each simulation step can be regarded as the microstructure that results from the deformation of a material with an unknown previous porosity to the actual situation. We refrain from any study of depth evolution of porosity, inclusion shape or distribution and remark, that the scope of the manuscript is a study of deformation and recrystallisation processes within the ice at the presence of a very weak phase.

Theoretically, the compaction of a pore is a function of the surface energy driving inward bubble surface movement and the inner bubble pressure counter-acting this movement. The latter depends on parameters such as the overburden pressure, bubble shape and connectivity. In a state of equilibrium, a bubble's size is does not change implying static conditions. Since the simulations do not incorporate gravitational forces, overburden pressure is unknown and the theoretical "area energy" is used to counter act surface energy (cf. section 2.4.1, equations (3) and (4) and Roessiger et al., 2014). The pre-factor $c$ can be regarded as an approximation of a compressibility factor that controls how quickly this equilibrium is reached (Roessiger et al., 2014). The lower the factor $c$, the less "area energy" is counter acting the surface energy that tends to decrease the overall cross sectional area of the bubbles. In turn, this means more cross sectional area change is allowed causing a stronger violation of the conservation of mass requirement.

To fulfil the conservation of mass requirement in our simulations, any movement of the ice-air interface that is not mass conserving should actually be inhibited. This would however lead to complete "freezing" of the interfaces, an even more unrealistic assumption. Therefore, we allow movements of the ice-air interfaces that preserve the overall porosity, but still allow for sufficient shape changes of the bubbles. Preparatory tests yielded $c = 0.1$ as a compromise to achieve this. With this, we use a 10 times higher factor $c$ than Roessiger et al. (2014), who modelled static conditions without deformation.

The current VPFFT code does, unfortunately, not include a compressible phase or voids. This is not an intrinsic limitation of the model, and a version without this limitation is under development. The current model is, therefore, not capable of simulating compaction, and we limited ourselves to area-conservative pure shear. Admittedly, this raises questions on the comparison of the simulations with the EDML firn image (Fig. 8). In the revised manuscript, we discuss the limitations associated with assuming incompressibility and explicitly highlight, that the comparison with the firn image has to be taken qualitatively and as a comparison of inferred processes and their expression in the microstructure.

Specific actions taken in the revised manuscript:

1.  As a reaction to both referees' concerns, we created the new sub-chapter "4.5 Limitations of the modelling approach" to discuss approximations made in our simulations. A condensed version of the explanations above is part of this chapter.

2.  The role of the pre-factor $c$ and our choice of $c = 0.1$ is now better explained in section 2.4.1 and in the new section 4.5.

3.  The comparison with the EDML firn image (Fig. 8) has moved to another new sub-chapter in discussion (section 4.2). We present the natural firn image as a first qualitative comparison with an Elle/VPFFT simulation on ice microdynamics. The intention of this comparison is trying to identify processes observed in the simulations also in natural firn (i.e. strain localisation in the vicinity of bubbles associated with enhanced dynamic recrystallisation). The limitations caused by the modelling approach are discussed and we state that we refrain from any quantitative comparison.

---

## Author Comment (AC4) · 11 Nov 2016

[revised manuscript text omitted]

**2.7.3 Microstructure mapping of natural samples**

To qualitatively compare the simulation results to natural firn microstructures, we utilized an image from a firn sample taken from the EPIA Dronning Maud Land ice core (EDML) site, Antarctica. The sample was cut from a core from 80 m depth. Using density and annual-layer thickness data of Kipfstuhl et al. (2009) for the EDML core, the density is about 800 kg/m³ at m depth and the total vertical shortening up to 50% (Faria et al., 2014b estimated this value based on the supplementary material from Ruth et al., 2007). A plane was cut from the sample and prepared for microstructure mapping (Kipfstuhl et al., 2009). The microstructure mapping technique allows detailed imaging of ice, air bubbles, (sub-) grain boundaries and other microstructural features typically found in polar ice. It is based on keeping a polished sample surface in a cold, yet dry environment, causing preferred ice sublimation at (sub-) grain boundaries. After sublimation, the prepared sample surface is scanned 
[revised manuscript text omitted]

~~Our numerically modelled microstructures resemble patterns observed in natural firn microstructures from EDML site (Fig. 8), supporting the microstructural evidence for the occurrence of dynamic recrystallisation in firn from the EDML ice core site (Kipfstuhl et al., 2009). However, the scale differs with the mean grain size in the models about one order of magnitude larger than in the EDML sample. It should be noted that the experimental strain rate (10⁻¹⁰ s⁻¹) is also about ten times larger than the 7.4·10⁻¹² s⁻¹ strain rate estimated for the sample (Faria et al., 2014b).~~

~~It should be noted that the strain rate in our simulations (10⁻¹⁰ s⁻¹) is an order of magnitude faster than assumed for the 80 m deep sample from the EDML core (Faria et al., 2014b). Modelling a slower strain rate is possible, but currently too time consuming. From a technical point of view, fast strain rates lead to numerical error reduction as the time steps for recrystallisation codes can be low. To achieve slower strain rates at the same time step, 
[revised manuscript text omitted]